# Fascin limits Myosin activity within *Drosophila* border cells to control substrate stiffness and promote migration

**Maureen C Lamb[1], Chathuri P Kaluarachchi[2], Thiranjeewa I Lansakara[2], Samuel Q Mellentine[1], Yiling Lan[2], Alexei V Tivanski[2], Tina L Tootle[1]\***

[1]Anatomy and Cell Biology, University of Iowa Carver College of Medicine, Iowa City, United States; [2]Department of Chemistry, University of Iowa, Iowa City, United States

**Abstract** A key regulator of collective cell migrations, which drive development and cancer metastasis, is substrate stiffness. Increased substrate stiffness promotes migration and is controlled by Myosin. Using *Drosophila* border cell migration as a model of collective cell migration, we identify, for the first time, that the actin bundling protein Fascin limits Myosin activity in vivo. Loss of Fascin results in: increased activated Myosin on the border cells and their substrate, the nurse cells; decreased border cell Myosin dynamics; and increased nurse cell stiffness as measured by atomic force microscopy. Reducing Myosin restores on-time border cell migration in *fascin* mutant follicles. Further, Fascin's actin bundling activity is required to limit Myosin activation. Surprisingly, we find that Fascin regulates Myosin activity in the border cells to control nurse cell stiffness to promote migration. Thus, these data shift the paradigm from a substrate stiffness-centric model of regulating migration, to uncover that collectively migrating cells play a critical role in controlling the mechanical properties of their substrate in order to promote their own migration. This understudied means of mechanical regulation of migration is likely conserved across contexts and organisms, as Fascin and Myosin are common regulators of cell migration.

**\*For correspondence:**
tina-tootle@uiowa.edu

**Competing interest:** The authors declare that no competing interests exist.

## Introduction

Cell migration is an essential process driving both development and cancer metastasis. During these processes, cells often migrate as groups or collectives, rather than single cells (*Friedl and Gilmour, 2009*). Collective cell migration requires that cell-cell adhesions be maintained amongst the cells to support cluster cohesion (*De Pascalis and Etienne-Manneville, 2017*). Additionally, many collective cell migrations occur in an invasive manner with the group of cells migrating between other cells or through basement membranes (*Chang et al., 2019*). During invasive migration, the environment puts mechanical forces on the migrating cells, causing them to respond by changing their shape and stiffness, and by modifying properties of their environment, such as extracellular matrix (ECM) composition (*Aguilar-Cuenca et al., 2014*; *Gasparski et al., 2017*; *Eble and Niland, 2019*). Therefore, stiffness has emerged as a critical regulator of collective cell migration.

During invasive, collective cell migration the group or cluster of cells must generate force necessary to invade through the ECM or other cells. Stiffness of the substrate is considered the primary regulator of the migrating cell's stiffness and ability to migrate (*Aguilar-Cuenca et al., 2014*). For example, increased substrate stiffness contributes to cancer cell migration and metastasis (*Gasparski et al., 2017*; *Oakes, 2018*; *Eble and Niland, 2019*). Indeed, hard matrices induce migration in breast cancer cells (*Ren et al., 2021*), and increased substrate stiffness promotes epithelial to mesenchymal

transitions (*Nieto and Cano, 2012*). While the role of substrate stiffness in promoting cell migration is well-established, most of these studies utilized in vitro culture systems. Therefore, it remains poorly understood how migrating cells are regulated in their native environments by the stiffness of their endogenous substrates.

A master regulator of cellular stiffness is Non-Muscle Myosin II (subsequently referred to as Myosin). Myosin is a force generating actin motor (*Vicente-Manzanares et al., 2009*; *Aguilar-Cuenca et al., 2014*). It is composed of two copies of three subunits: two heavy chains, two essential light chains, and two regulatory light chains (MRLC; *Vicente-Manzanares et al., 2009*; *Aguilar-Cuenca et al., 2014*). Myosin activation is regulated through phosphorylation of its regulatory light chains. This phosphorylation occurs through a number of kinases, including Myosin light chain kinase (MLCK) and Rho-associated kinase (Rok), and dephosphorylation occurs through phosphatases, such as protein phosphatase 1 c (PP1c) and its catalytic subunit, Myosin binding subunit (Mbs *Vicente-Manzanares et al., 2009*; *Aguilar-Cuenca et al., 2014*). Myosin generates cortical tension by associating with and acting upon cortical F-actin; this regulates cell stiffness which can influence cell migration (*Butcher et al., 2009*; *Aguilar-Cuenca et al., 2014*). Importantly, Myosin regulates stiffness in both cellular substrates and migrating cells during many different cell migrations (*Lo et al., 2000*; *Vicente-Manzanares et al., 2009*; *Mohan et al., 2015*). Additionally, Myosin not only generates mechanical force within a cell but aids in sensing and responding to external forces applied to the cell (*Butcher et al., 2009*; *Vicente-Manzanares et al., 2009*; *Aguilar-Cuenca et al., 2014*).

A recently discovered regulator of Myosin is Fascin. Fascin is an F-actin binding protein that bundles or cross-links actin filaments into fibers (*Jayo and Parsons, 2010*; *Hashimoto et al., 2011*). However, recent studies demonstrate that there are many non-canonical roles for Fascin (*Lamb and Tootle, 2020*). One of these non-canonical functions of Fascin is the regulation of Myosin (*Elkhatib et al., 2014*). Increasing concentrations of Fascin in an in vitro system decreased Myosin ATP consumption and motor speed along actin filaments (*Elkhatib et al., 2014*). These data suggest that Fascin limits Myosin activity (*Elkhatib et al., 2014*). Whether Fascin limits Myosin activity to control substrate stiffness and thereby cell migration remains unknown. Notably, Fascin has well-established roles in promoting cell migration (*Lamb and Tootle, 2020*). Fascin aids in the formation of cell migratory structures like filopodia (*Hashimoto et al., 2011*) and invadopodia (*Li et al., 2010*). Fascin promotes many types of cell migrations in development and disease, including cancer metastasis (*Ma and Machesky, 2015*). Investigation of Fascin's role in promoting cell migration has primarily focused on Fascin as an F-actin bundler and it is unknown if Fascin limits Myosin activity to regulate collective cell migration.

An ideal model to uncover the role of Fascin in regulating Myosin during collective cell migration in a native context is *Drosophila* border cell migration. Border cell migration occurs during Stage 9 (S9) of oogenesis. During S9, the follicle is composed of an oocyte and 15 germline-derived nurse cells that are surrounded by a layer of somatic epithelial cells called follicle cells (*Spradling, 1993*). Surrounding the follicle cells is a layer of ECM that envelopes the follicle (*Spradling, 1993*). Inside the follicle, however, there is limited evidence of any ECM (*Medioni and Noselli, 2005*). At the beginning of S9, a group of 8–10 follicle cells are specified as border cells and delaminate from the epithelium to start their migration (*Montell, 2003*). The border cells migrate invasively and collectively between the nurse cells until they reach the nurse cell-oocyte boundary (*Figure 1A and B*; *Montell, 2003*). Border cell migration is a cell-on-cell migration in which the nurse cells are the substrate for the migration, there is only a small puncta of ECM on the border cell cluster as it migrates (*Medioni and Noselli, 2005*) and border cell migration is largely independent of Integrin-based adhesions (*Dinkins et al., 2008*; *Llense and Martín-Blanco, 2008*). Importantly, similar to other types of migration, the stiffness of the nurse cell substrate regulates both the stiffness of the border cells and their migration (*Aranjuez et al., 2016*). Therefore, border cell migration is a powerful model for studying invasive, collective cell migration as the cluster of migrating cells can be visualized in its native context using both fixed and live imaging. Additionally, the factors that regulate border cell migration play conserved roles in other invasive, collective cell migrations, including cancer metastasis (*Montell et al., 2012*; *Stuelten et al., 2018*). Indeed, both Fascin and Myosin play roles in promoting cancer metastasis (*Hashimoto et al., 2011*; *Aguilar-Cuenca et al., 2014*) and on-time border cell migration (*Figure 1C*, *Edwards and Kiehart, 1996*; *Lamb et al., 2020*). We previously found that Fascin (*Drosophila* Singed, Sn) is required for both border cell delamination and proper protrusion localization (*Lamb et al., 2020*). Both loss and activation of Myosin result in similar phenotypes of delayed delamination and mislocalized

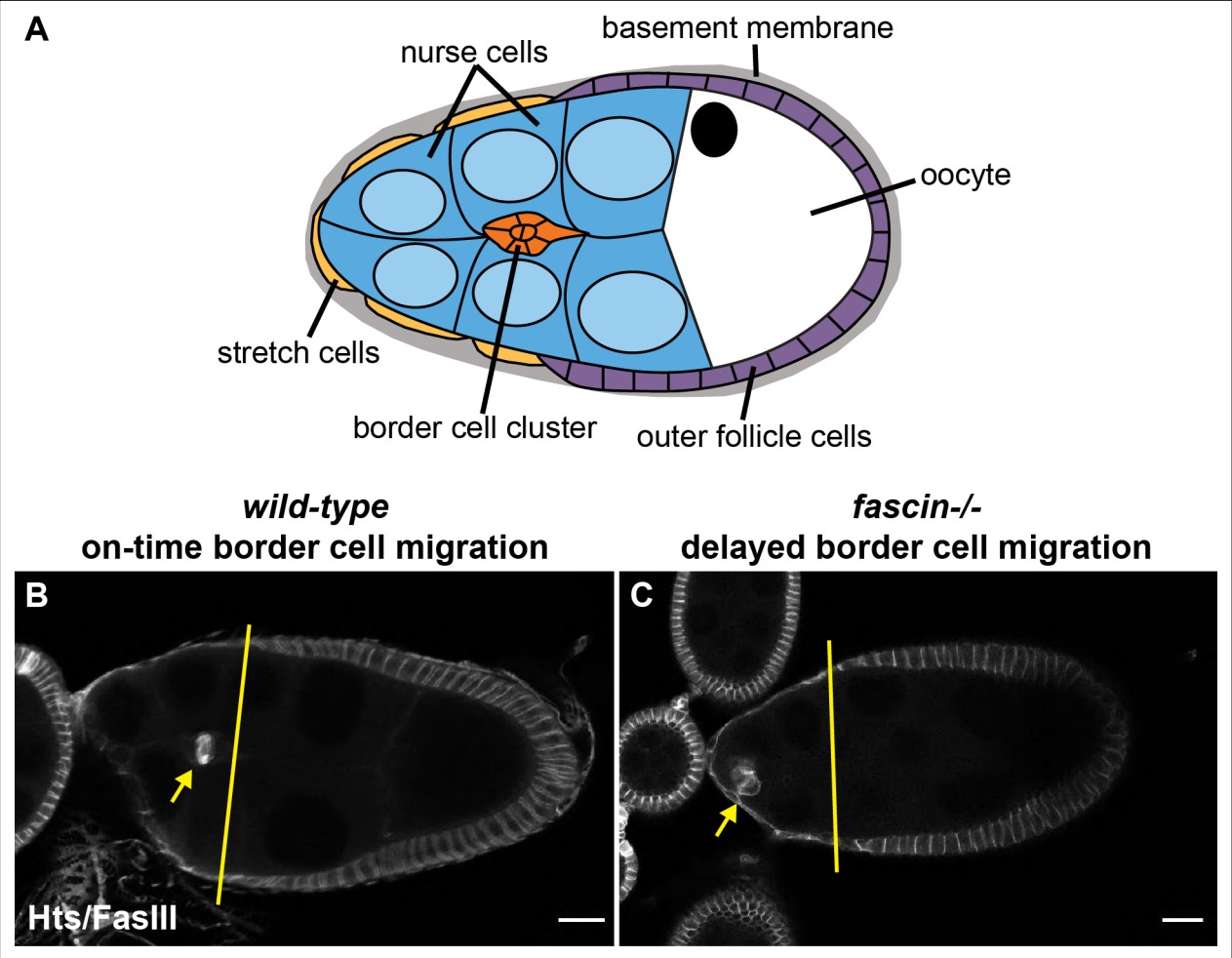

**Figure 1.** Fascin is required for on-time border cell migration during Stage 9. (**A**) Schematic of a Stage 9 *Drosophila* follicle. The nurse cells (blue) are the substrate for the migrating border cell cluster (orange); the direction of border cell migration is to the right. The follicle is surrounded by a layer of somatic epithelial cells which include the outer follicle cells (purple) and the stretch cells (gold). These cells are surrounded by a layer of basement membrane (grey). (**B–C**). Maximum projections of 2–4 confocal slices of Stage 9 follicles of the indicated genotypes. Border cell stain (Hts/FasIII, white). (**B**) wild-type (*yw*). (**C**) *fascin*-null (*fascin^{sn28/sn28}*). Yellow lines indicate the position of the outer follicle cells and the yellow arrows indicate the position of the border cell cluster. In wild-type follicles, the border cells are in line with the position of the outer follicle cells (**B**), whereas in *fascin* mutant follicles the border cells are significantly behind the outer follicle cells, indicating that loss of Fascin results in delayed border cell migration during Stage 9 of oogenesis (**C**). All genotypes are listed in Table 1.

border cell cluster protrusions (*Majumder et al., 2012*; *Aranjuez et al., 2016*; *Mishra et al., 2019*). These data suggest that the cycling of Myosin between active and inactive forms controls border cell migration. Thus, border cell migration is an ideal system to uncover the relationship of Fascin and Myosin during collective cell migration.

Here, we demonstrate for the first time that Fascin inhibits Myosin activity in vivo. Loss of Fascin significantly increases the level of active Myosin, reduces Myosin dynamics, and increases nurse cell (aka substrate) stiffness as quantified by atomic force microscopy (AFM) nanoindentation technique. Reducing Myosin in *fascin* mutant follicles rescues border cell migration delays, indicating that Fascin's tight regulation of Myosin activity is critical for on-time migration. Further, a phosphomimetic form of Fascin that precludes F-actin bundling is unable to limit Myosin activation, supporting the prior model that Fascin limits Myosin activity by tightly bundling F-actin and preventing Myosin binding to actin filaments (*Elkhatib et al., 2014*). We used RNAi knockdown and rescue experiments to assess the cell-specific roles of Fascin in regulating Myosin activity and nurse cell stiffness. Based on the literature, we expected that Fascin would primarily function within the nurse cells to control both substrate stiffness and Myosin activity within both the nurse cells and border cells. Surprisingly, we find that

knowing down Fascin in the border cells increases the level of active Myosin on both the border cells and the nurse cells, and increases the stiffness of the nurse cells. Similarly, re-expressing Fascin in only the border cells of *fascin* mutants restores normal Myosin activity levels and stiffness of the nurse cells. These unexpected findings suggest that migrating cells influence the mechanobiology of their substrate to promote their migration. Supporting this, increasing Rok activity in the border cells also results in increased nurse cell stiffness, indicating this migratory cell regulation of substrate stiffness is not a Fascin-specific phenomenon. Together these findings lead to the following model: Fascin acts primarily within the migrating border cells to limit Myosin activation which controls the stiffness of the both the border cells and their substrate, the nurse cells, to promote on-time migration. It is likely that this regulation of Myosin by Fascin and thereby, migrating cells controlling substrate stiffness, is a conserved means of promoting collective cell migration.

## Results

### Fascin inhibits Myosin activation in the *Drosophila* follicle

Previous data demonstrates that Fascin can inhibit the activity of Myosin in vitro (*Elkhatib et al., 2014*). Fascin functions in both the nurse cells and the border cells to promote on-time border cell migration (*Lamb et al., 2020*). Additionally, during border cell migration Myosin generates forces in the nurse cells that push upon the border cells, causing the border cells to activate Myosin and stiffen (*Aranjuez et al., 2016*), suggesting that the nurse cells control the stiffness of the border cell cluster. Based on these observations, we hypothesized that Fascin may regulate Myosin activity in the *Drosophila* follicle, specifically the nurse cells, to promote border cell migration.

To test this hypothesis, we assessed if Fascin limits Myosin activity in the *Drosophila* follicle. Myosin is activated via phosphorylation on its regulatory light chain subunit (MRLC). To assess changes in Myosin activation in the follicle, we stained follicles using an antibody against phosphorylated MRLC (pMRLC); wild-type and *fascin*-null follicles were stained in the same tube to account for staining variability. We observe a striking increase in active MRLC along both the nurse cell and border cell membranes of *fascin*-null follicles (*Figure 2B* compared to A, blue arrows and B' compared to A', orange arrows). We quantified levels of active MRLC by measuring the relative fluorescence intensity of pMRLC on the nurse cell and border cell membranes (*Figure 2C–D*; for example quantifications see *Figure 2—figure supplement 1A,B*). Briefly, for the nurse cell quantifications, 3 line segments per follicle were drawn across nurse cell-nurse cell membranes and the fluorescence intensity peak for pMRLC was normalized to phalloidin intensity at the same point; phalloidin intensity is not affected by loss of Fascin (*Figure 2—figure supplement 1C*). Concurrent border cell staining ensured we did not measure across a border cell cluster protrusion. The three values were then averaged for a single image (for example quantifications see *Figure 2—figure supplement 1A,B*). For border cell intensity, the border cell cluster was traced using the phalloidin or border cell stain and the mean fluorescence intensity for pMRLC was measured and normalized to the mean fluorescence intensity of pMRLC of the same shape in the nurse cell cytoplasm (for example quantifications see *Figure 2—figure supplement 1A*). We used the nurse cell cytoplasm pMRLC stain as the background for the border cell measurement because there is no available antibody that works against MRLC. Importantly, nurse cell cytoplasmic pMRLC intensity (*Figure 2—figure supplement 1D*) and Myosin heavy chain (*Drosophila* Zipper) protein levels are not significantly different between wild-type and *fascin*-null follicles (*Figure 2—figure supplement 1E,F*). For increased clarity, throughout the entire manuscript, all graphs quantifying MRLC activity on the nurse cell membranes are shown using blue circles and MRLC activity on the border cell cluster are shown using orange circles. We find that there is a significant increase in active MRLC intensity on the *fascin*-null nurse cell membranes compared to wild-type follicles (*Figure 2C*, p < 0.0001). Additionally, active MRLC is also significantly increased on the border cell cluster when Fascin is lost (*Figure 2D*, p < 0.0001). Further, we assessed the spatial distribution of active MRLC on the nurse cell membranes surrounding the border cell cluster. In both wild-type and *fascin* mutant follicles, we observe active Myosin enriched on nurse cell membranes in front, behind, and on the sides of the migrating cluster (*Figure 2A and B*). While the intensity of pMRLC staining is higher in the *fascin* mutants indicating higher Myosin activation, there does not seem to be a change in the spatial distribution of active Myosin on the border cell cluster.

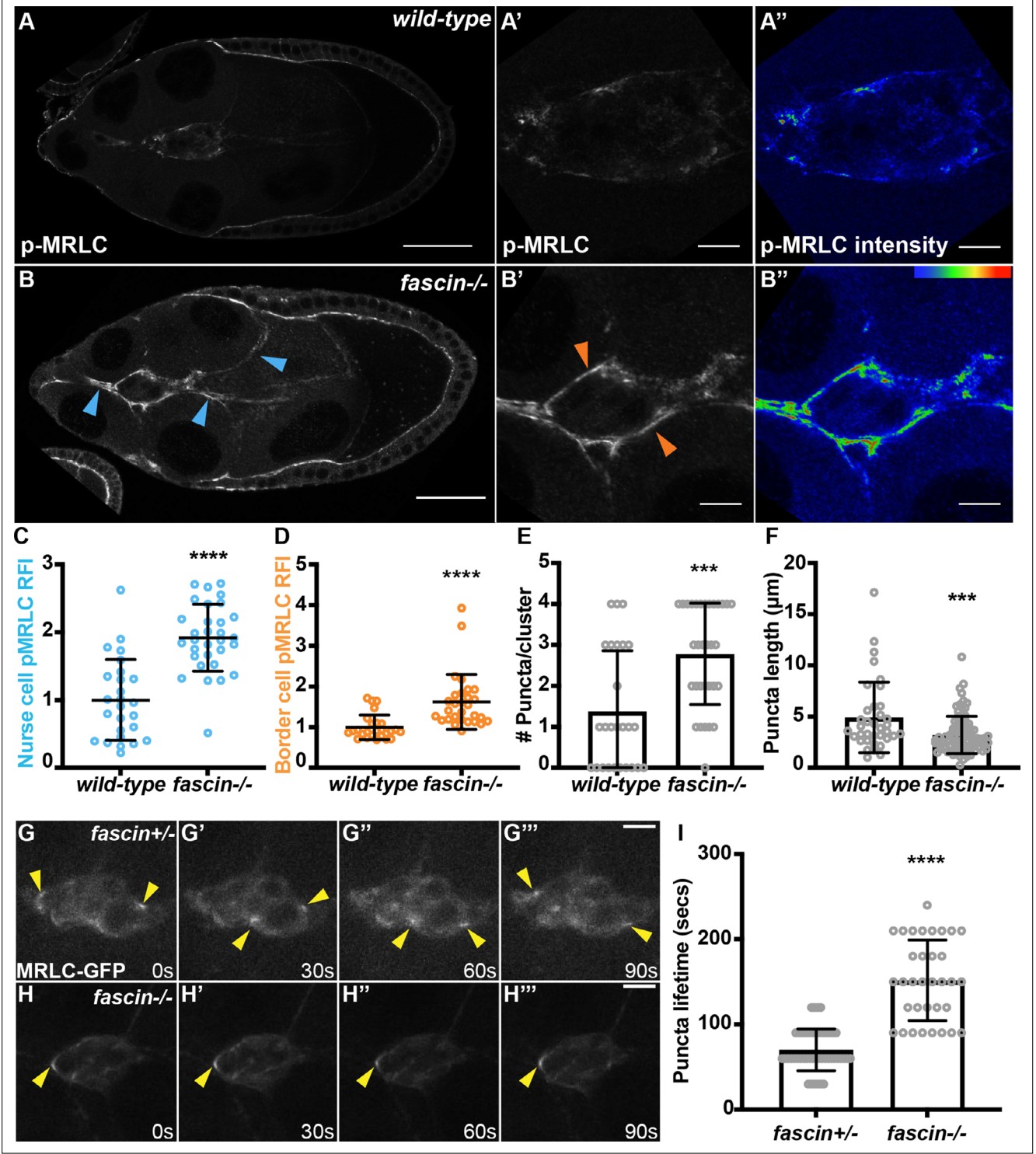

**Figure 2.** Fascin limits Myosin activity in the Stage 9 *Drosophila* follicle. (**A-B″**) Maximum projections of 2–4 confocal slices of Stage 9 follicles of the indicated genotypes. (**A-A′, B-B′**) phospho-MRLC (pMRLC, white). (**A″, B″**) pMRLC pseudocolored with Rainbow RGB, red = highest intensity pixels. (**A-A″**) wild-type (*yw*). (**B-B″**) *fascin*-null (*fascin^sn28/sn28*). Samples were stained in the same tube. Blue arrows = pMRLC enrichment on nurse cells. Orange arrows = pMRLC enrichment on border cell cluster. Scale bars = 50 μm in **A, B** and 10 μm in **A′-A″, B′-B″**. (**C–F**) Graphs of quantification of pMRLC intensity and localization at the nurse cell membranes (**C**) and border cell cluster (**D, E, F**) in wild-type and *fascin*-null follicles. Each circle represents a follicle. Error bars = SD. ***p < 0.001, ****p < 0.0001 (unpaired t-test). In **C**, peak pMRLC intensity was quantified at the nurse cell membranes and normalized to phalloidin staining in the same follicle, three measurements were taken per follicle and averaged. In **D**, pMRLC intensity on the border cell cluster was quantified and normalized to background pMRLC staining in the same follicle. For examples of the quantifications in **C and D** see

*Figure 2 continued*

*Figure 2—figure supplement 1*. In **E**, the number of Myosin puncta per cluster was manually counted. In **F**, the maximum length of each Myosin puncta was measured. (**G-H‴**) Maximum projection of three confocal slices from time-lapse imaging of MRLC-GFP in the indicated genotypes. Direction of migration is to the right. Scale bars = 10 μm. (**G-G‴**) Control follicle (*fascin^{sn28}/+; MRLC-GFP/+*; ***Video 1***). (**H-H‴**) *fascin*-null follicle (*fascin^{sn28/sn28}; MRLC-GFP/+*; ***Video 2***). (**I**) Quantification of puncta lifetime from time-lapse imaging for control (n = 4) and *fascin*-null (n = 4) MRLC-GFP expressing follicles. Puncta lifetime was defined as the amount of time elapsed from when a punctum first appeared to when it completely disappeared. ****p < 0.0001 (unpaired t-test). Error bars = SD. *fascin*-null follicles have increased pMRLC on the the nurse cell membranes (**B, C**) and border cell cluster (**B′, D**) compared to wild-type follicles (**A, A′, C, D**). The border cell clusters in *fascin*-null mutants also have increased Myosin puncta number but decreased length (**E, F**). *fascin* mutants have significantly slowed Myosin dynamics (**H-H‴, I**) compared to the control clusters (**G-G‴, I**). All genotypes are listed in Table 1.

The online version of this article includes the following source data and figure supplement(s) for figure 2:

**Source data 1.** Source data for *Figure 2C-F and I*.

**Figure supplement 1.** Myosin activity assessments.

**Figure supplement 1—source data 1.** Source data for *Figure 2—figure supplement 1C, D and F*.

**Figure supplement 1—source data 2.** Source data for *Figure 2—figure supplement 1E*.

We also quantified changes in active MRLC puncta number and length on the border cell cluster (***Figure 2E and F***; see Methods for quantification details). Briefly, the puncta on each border cell cluster were manually counted and the maximum length of each punctum was measured from maximum projections of 2–4 confocal slices using ImageJ software. Loss of Fascin increases puncta number but decreases puncta length (***Figure 2E and F***, p < 0.001). Together these results demonstrate that Fascin limits Myosin activation in the *Drosophila* S9 follicle on both the nurse cell membranes and the border cell cluster, providing the first evidence that Fascin regulates Myosin activity in vivo.

## Fascin limits Myosin dynamics on the migrating border cell cluster

We next wanted to determine how Fascin influences Myosin dynamics during border cell migration. In addition to the level of activation, the localization and dynamics of Myosin influence invasive migration (***Vicente-Manzanares et al., 2009***; ***Majumder et al., 2012***; ***Aguilar-Cuenca et al., 2014***; ***Aranjuez et al., 2016***). Indeed, during border cell migration, dynamic cycles of Myosin activation and inactivation at the cluster membrane are essential for proper migration (***Aranjuez et al., 2016***). We visualized Myosin dynamics on the border cell cluster using a C-terminally GFP-tagged MRLC (MRLC-GFP; *Drosophila* Spaghetti Squash, Sqh), under the control of its endogenous promoter. Previous data demonstrates that MRLC-GFP is highly expressed on the border cell cluster during migration and accumulates in transient puncta on the cluster; these puncta depend on Myosin activation, suggesting they are sites of active Myosin (***Majumder et al., 2012***). Using live imaging, we find in control follicles, MRLC-GFP puncta appear and disappear rapidly on the border cell cluster (***Figure 2G–G‴***, ***Video 1***). However, in the *fascin*-null follicles, the MRLC-GFP puncta dynamics are much slower (***Figure 2H–H‴***, ***Video 2***). We quantified this change in MRLC-GFP dynamics by measuring puncta lifetime on the cluster (***Figure 2I***). The control follicles display an average puncta lifetime of 70.2 s, while in *fascin*-null follicles the average puncta lifetime is 151.8 s (***Figure 2I***, p < 0.0001). These results suggest that Fascin limits Myosin dynamics on the migrating border cell cluster.

## Fascin regulates nurse cell stiffness

As increased Myosin activity increases actomyosin contractility and cell stiffness, we next wanted to directly measure the stiffness of *fascin*-null follicles. Substrate stiffness is thought to be a driving regulator of cell migration and migrating cell stiffness

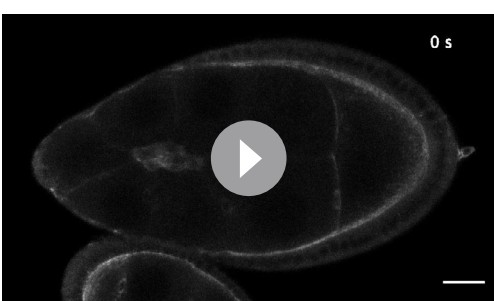

**Video 1.** Myosin dynamics in control follicle. Video of Stage 9 control MRLC-GFP expressing follicle (*fascin^{sn28}/+; MRLC-GFP/+*). Time listed in seconds. Images were acquired every 30 s. Anterior is to the right. Scale bar = 20 μm. The control cluster displays Myosin dynamics in which Myosin puncta appear and disappear rapidly on the border cell cluster. All genotypes are listed in ***Table 1***.

https://elifesciences.org/articles/69836/figures#video1

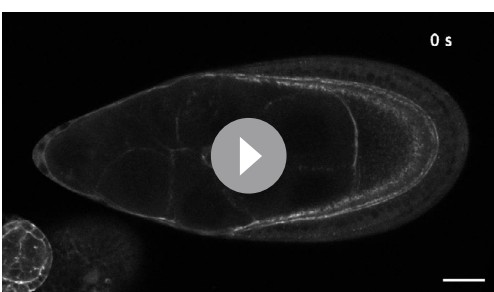

**Video 2.** Myosin dynamics in *fascin*-null follicle. Video of Stage 9 *fascin*-null MRLC-GFP expressing follicle (*fascin*^sn28/sn28; *MRLC-GFP/+*). Time listed in seconds. Images were acquired every 30 s. Anterior is to the right. Scale bar = 20 µm. Loss of Fascin results in slower Myosin dynamics, with Myosin puncta appearing and remaining longer than in controls (see Video 1). All genotypes are listed in *Table 1*.
https://elifesciences.org/articles/69836/figures#video2

(*Di Martino et al., 2016*; *Gasparski et al., 2017*; *Oakes, 2018*; *Ren et al., 2021*), therefore we aimed to directly quantify nurse cell stiffness. AFM is a standard method to directly measure mechanical properties of biological tissues (*Kreplak, 2016*). AFM can be used to quantify the elastic modulus, which is a measurement of how easily an elastic material is deformed when a known amount of force is applied (*Kreplak, 2016*). A high elastic modulus value corresponds to a stiff tissue. We used AFM nanoindentation technique to quantify the stiffness of *fascin*-null and wild-type nurse cells (*Crest et al., 2017*; *Chen et al., 2019*). The nurse cells are the substrate for the border cells and their stiffness regulates border cell migration and cluster stiffness (*Aranjuez et al., 2016*). Notably, during S9, the nurse cells are surrounded by a layer of stretch follicle cells and a basement membrane that envelopes the entire follicle (*Figure 3A*). Previous measurements on *Drosophila* follicles using AFM established that there is significant difference in stiffness between the basement membrane and the underlying nurse cells (*Crest et al., 2017*; *Chen et al., 2019*). These different tissues stiffnesses can be separated by using different indentation ranges to indent the AFM probe into just the basement membrane or to indent deeper into the nurse cells (*Figure 3B*; *Chlasta et al., 2017*). Thus, by using two indentation ranges to fit the mechanical response, we can quantify the distinct stiffness of the basement membrane versus that of the underlying nurse cells (*Chlasta et al., 2017*).

We used AFM and the Hertzian elastic contact model to calculate the stiffness of wild-type and *fascin*-null S9 follicles (*Figure 3C*). For increased clarity, throughout the entire manuscript all graphs quantifying stiffness by AFM are represented using green circles. For an indentation range of 0–100 nm, which probes the basement membrane, wild-type follicles have an average stiffness of 24.2 kPa and *fascin*-null follicles have a similar average stiffness of 26.5 kPa (*Figure 3D*, p > 0.05). However, for an indentation range of 310–550 nm, which probes the nurse cell stiffness, wild-type follicles have an average stiffness of 10.1 kPa while *fascin*-null follicles have a significantly increased average stiffness of 25.9 kPa (*Figure 3D*, p < 0.0001). Thus, the stiffness of the *fascin*-null nurse cells is >2 x higher than wild-type nurse cells. Together these results demonstrate that loss of Fascin increases the stiffness of the nurse cells in S9 *Drosophila* follicles.

## Fascin limits Myosin activity to promote border cell migration

As increased stiffness of the nurse cells or border cells inhibits border cell migration (*Aranjuez et al., 2016*), we hypothesized that the increased Myosin activity in *fascin*-null follicles contributes to the previously characterized border cell migration delays (*Lamb et al., 2020*). To address this hypothesis, we first used a pharmacological inhibitor of Myosin and assessed the effect on border cell migration. Follicles were incubated for 2 hr in either control media or 200 µM of Y-27632, a Rho inhibitor previously used to reduce Myosin activity in *Drosophila* follicles (*He et al., 2010*). This inhibitor reduces activated Myosin levels on both the nurse cells and border cells (*Figure 4—figure supplement 1A,B*). We then employed our previously developed method to quantify delays in border cell migration during S9, which takes the ratio of the distance the border cells have migrated from the anterior end of the follicle to the distance of the outer follicle cells from the anterior end of the follicle (see schematic *Figure 4A*; *Lamb et al., 2020*). We call this value the migration index; for increase clarity, throughout the entire manuscript all migration indexes data are shown in magenta. A migration index of approximately one indicates on-time migration during S9, while a value less that one indicates delayed migration and a value greater than one indicates an accelerated migration. As we previously established, loss of Fascin significant delays migration (*Figure 1C*, *Lamb et al., 2020*). Here, we find that inhibiting Myosin activity with Y-27632 in *fascin*-null follicles restores on-time border cell

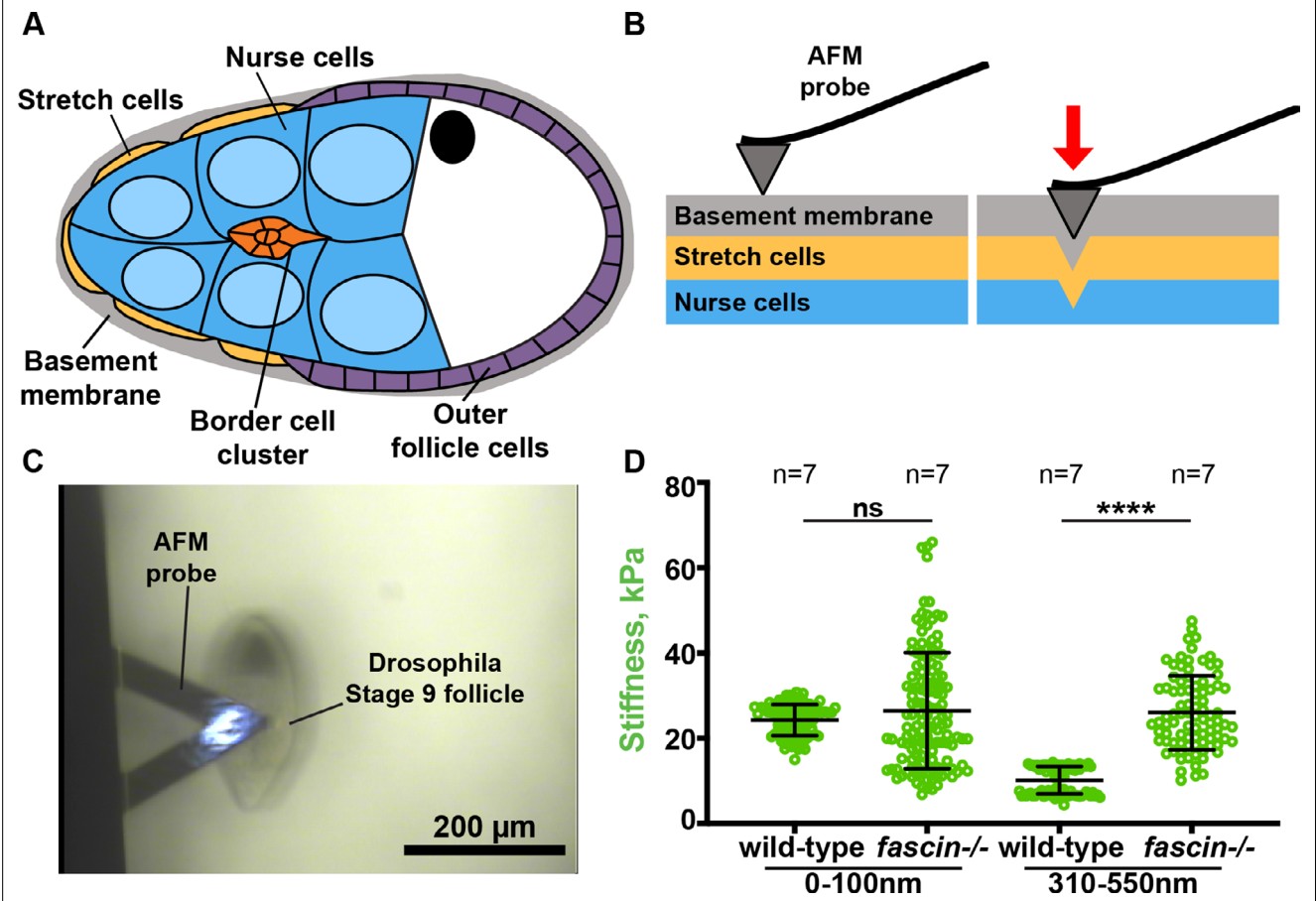

**Figure 3.** Fascin regulates nurse cell stiffness in the *Drosophila* follicle. (**A**) Schematic of Stage 9 *Drosophila* follicle. The nurse cells (blue) are surrounded by a layer of stretch cells (gold) and basement membrane (gray). (**B**) Schematic of AFM probe indentation through the basement membrane (gray) and stretch cells (gold) into the underlying nurse cells (blue). (**C**) Bright-field image of AFM probe over a Stage 9 follicle. (**D**) Graph of nurse cell stiffness (kPa) in wild-type or *fascin*-null follicles as measured by AFM. Each circle represents a single indentation. Error bars = SD. ns indicates p > 0.05, ****p < 0.0001 (unpaired t-test). Loss of Fascin significantly increases the stiffness of the nurse cells (**D**). All genotypes are listed in Table 1.

The online version of this article includes the following source data for figure 3:

**Source data 1.** Source data *Figure 3F*.

migration compared to the *fascin*-null control (*Figure 4C–D and G*, migration index 1.1 compared to 0.78) and is not significantly different from the wild-type control (*Figure 4B and G*, migration index 1.1 compared to 0.95).

As loss of Fascin increases Myosin activity on both the nurse cells and the border cells, we next sought to determine whether increased Myosin activity on the nurse cells and/or border cells is responsible for delays in border cell migration. We used the UAS/GAL4 system to express an RNAi against MRLC (*Drosophila* Sqh) to knockdown Myosin in *fascin*-null mutants in different cell types (see schematic of cell specific knockdown in Figure 6A) – the germline (*matα* GAL4), somatic (*c355* GAL4), or border cells (*c306* GAL4). Unfortunately, knockdown of Myosin in the somatic (*c355* GAL4) or border cells (*c306* GAL4) was lethal, however knockdown of Myosin in the germline (*matα* GAL4) was viable. Germline knockdown of MRLC in *fascin* mutants significantly decreased active Myosin levels on the nurse cells compared to *fascin*-null controls (*Figure 4—figure supplement 1C*). However, it fails to restore normal levels of active Myosin on the border cell cluster, as Myosin activation remains significantly increased compared to the wild-type control and is not signficantly different than the *fascin*-null control (*Figure 4—figure supplement 1D*). We next assessed whether altering Myosin activity within the nurse cells can restore on-time border cell migration in *fascin*-null mutants using the migration index quantification (see schematic, *Figure 4A*). Germline knockdown of MRLC in *fascin* mutant follicles rescues border cell migration (*Figure 4E, F and H*, migration index 0.91 compared to

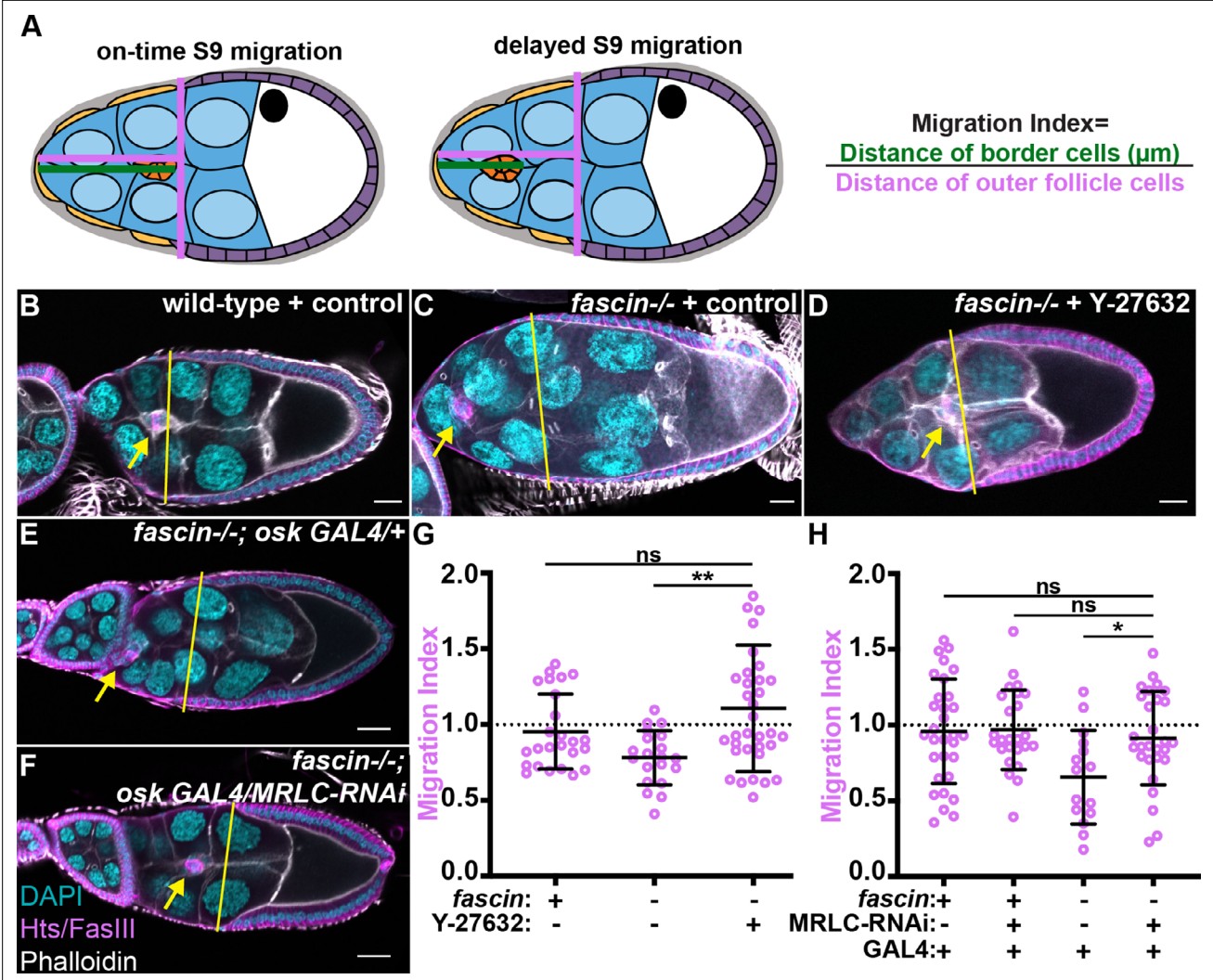

**Figure 4.** Reducing Myosin activity rescues border cell migration in *fascin* mutant follicles. (**A**) Schematics of the migration index quantification for on-time and delayed border cell migration during Stage 9. The migration index is the distance the border cell cluster has migrated (green line) divided by the distance of the outer follicle cells from the anterior end of the follicle (magenta line). A value of ~1 indicates on-time migration, a value <1 indicates delayed migration and a value >1 indicates accelerated migration. (**B–F**) Maximum projections of 2–4 confocal slices of Stage 9 follicles of the indicated genotypes. Merged images: Hts/FasIII (magenta, border cell migration stain), phalloidin (white), and DAPI (cyan). Yellow lines = outer follicle cell distance. Yellow arrows = border cell cluster. Black boxes have been added behind text. Scale bars = 20 μm. (**B**) wild-type (*yw*) treated with control S9 media + vehicle (DMSO). (**C**) *fascin-/-* (*fascin^sn28/sn28*) treated with control S9 media + vehicle. (**D**) *fascin-/-* (*fascin^sn28/sn28*) treated with 200 μM of Y-27632. (**E**) *fascin^sn28/sn28; oskar GAL4 (2)/+* (**F**) *fascin^sn28/sn28; oskar GAL4 (2)/MRLC-RNAi*. (**G, H**) Migration index quantification of the indicated genotypes. Dotted line at 1 = on time migration. Circle = Stage 9 follicle. Lines = averages and error bars = SD. ns indicates p > 0.05, *p< 0.05, **p < 0.01 (one-way ANOVA with Tukey's multiple comparison test). Pharmacological inhibition of Myosin activity rescues border cell migration delays in *fascin* mutant follicles (**B–D, G**). Similarly, germline knockdown of MRLC restores on-time border cell migration in *fascin* mutants, suggesting that increased active Myosin in the nurse cells of *fascin* mutants leads to the border cell migration delays (**E, F, H**). All genotypes are listed in Table 1.

The online version of this article includes the following source data and figure supplement(s) for figure 4:

**Source data 1.** Source data for *Figure 4G, H*.

**Figure supplement 1.** Pharmacological inhibition of Myosin and germline MRLC knockdown reduce active Myosin in the follicle.

**Figure supplement 1—source data 1.** Source data for *Figure 4—figure supplement 1A-D*.

0.65). Together these results suggest that Fascin is required to limit Myosin activity within the nurse cells to promote on-time border cell migration.

## Phosphorylation of Fascin controls its ability to limit Myosin activity

Previous data demonstrated that in vitro Fascin can limit Myosin activation; however, the mechanism of how Fascin regulates Myosin activity is unknown (*Elkhatib et al., 2014*). It was hypothesized that Fascin's ability to tightly bundle F-actin precludes Myosin from being able to bind to actin filaments and generate force (*Elkhatib et al., 2014*). Phosphorylation of Fascin at serine 52 (S52, mammalian S39) inhibits its F-actin bundling function (*Yamakita et al., 1996*; *Ono et al., 1997*). If Fascin's bundling activity is required to limit Myosin activation, we would predict that global expression (*actin 5* c GAL4) of phosphomimetic Fascin (S52E) in *fascin*-null mutants would fail to suppress the increased active

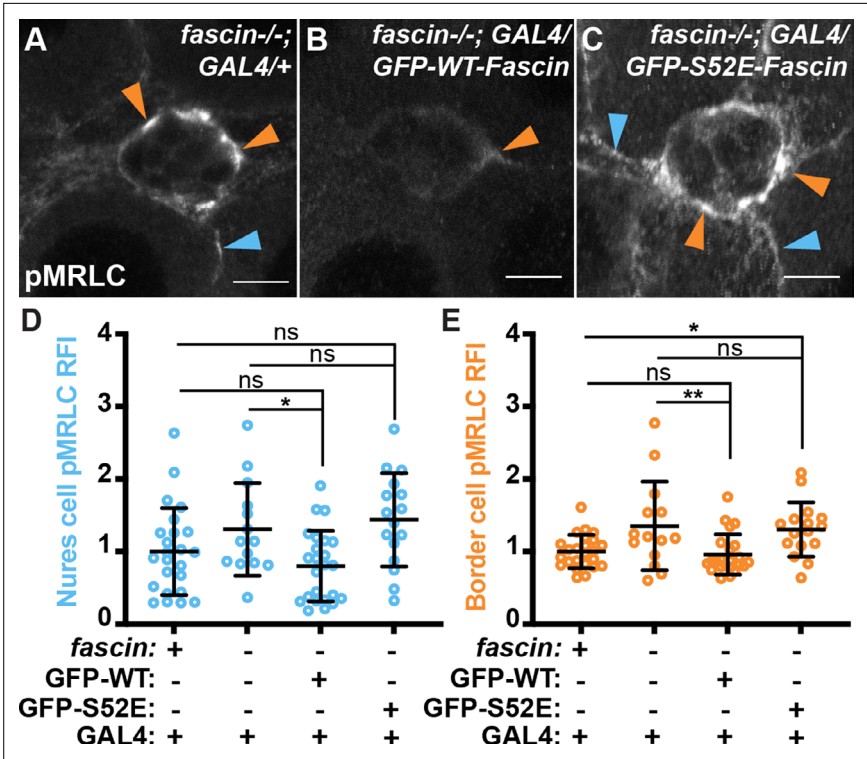

**Figure 5.** Phosphorylated Fascin fails to limit Myosin activation. (**A–C**) Maximum projections of 2–4 confocal slices of Stage 9 follicles of the indicated genotypes stained for phospho-MRLC (pMRLC, white). Blue arrows = pMRLC enrichment on surrounding nurse cells. Orange arrows = pMRLC enrichment on border cell cluster. Scale bars = 10 μm. (**A**) *fascin* mutant with global GAL4 (*fascin^{sn28/sn28}*; *actin5c GAL4/+*). (**B**) Global GFP-Fascin expression in *fascin* mutant (*fascin^{sn28/sn28}*; *actin5c GAL4/UAS-GFP-Fascin*). (**C**) Global GFP-Fascin-S52E expression in *fascin* mutant (*fascin^{sn28/sn28}*; *actin5c GAL4/UAS-GFP-Fascin-S52E*). (**D, E**) Graphs of quantification of pMRLC intensity at the nurse cell membranes (**D**) and border cell cluster (**E**) in the indicated genotypes. Each circle represents a follicle. Error bars = SD. ns indicates p > 0.05, *p < 0.05, **p < 0.01 (One-way ANOVA with Tukey's multiple comparison test). In **D**, peak pMRLC intensity was quantified at the nurse cell membranes and normalized to phalloidin staining in the same follicle, three measurements were taken per follicle and averaged. In **E**, pMRLC intensity on the border cell cluster was quantified and normalized to background staining in the same follicle. Restoring wild-type Fascin expression in both the somatic and germline cells of *fascin* mutant follicles (**B**) significantly reduces activated Myosin enrichment on the nurse cell membranes (**D**) and border cell cluster (**E**) compared to the *fascin*-null control (**B, D, E**). Whereas expressing a phosphomimetic form of Fascin in a *fascin* mutant (**C**) does not alter activated Myosin on the nurse cell membranes (**D**) or border cell cluster (**E**). All genotypes are listed in Table 1.

The online version of this article includes the following source data and figure supplement(s) for figure 5:

**Source data 1.** Source data for *Figure 5D, E*.

**Figure supplement 1.** Phosphorylation of Fascin regulates border cell migration.

**Figure supplement 1—source data 1.** Source data *Figure 5—figure supplement 1C*.

Myosin. As a control, we find that global expression of wild-type Fascin (GFP-Fascin) in *fascin* mutant follicles significantly reduces active MRLC enrichment on both the nurse cell membranes (*Figure 5*) and border cell cluster (*Figure 5B*, orange arrow and E). Conversely, when the phosphomimetic form of Fascin (GFP-Fascin S52E) is expressed in *fascin*-null mutants, we observe high levels of active Myosin on both the nurse cell membranes (*Figure 5C*, blue arrows and D) and border cell cluster (*Figure 5C*, orange arrows and E) that are not significantly different than the *fascin* mutant control (*Figure 5A, D and E*). These data support the model that Fascin limits Myosin activity by bundling F-actin and precluding Myosin's ability to bind to actin filaments.

As we found that tight regulation of Myosin activity by Fascin is critical for on-time border cell migration (*Figure 4*), and expression of phosphomimetic Fascin (S52E) in *fascin* mutant follicles fails to restore normal levels of Myosin activity (*Figure 5B–E*), we expected it would also fail to fully rescue the delays in border cell migration. We previously found global expression (*actin 5 c GAL4*) of wild-type Fascin in *fascin* mutant follicles restores on-time border cell migration (*Lamb et al., 2020*). As expected, when we quantify the migration index (described in *Figure 4A*) for *fascin* mutant follicles with global expression of phosphomimetic Fascin (S52E), we find it only partially rescues delays in border cell migration (*Figure 5—figure supplement 1C*, migration index 0.90 compared to 0.80). Together these data indicate Fascin functions in other ways besides bundling F-actin and limiting Myosin activity to promote on-time border cell migration.

## Fascin acts in the border cells to control substrate stiffness

Previous evidence demonstrated that the stiffness of the nurse cells regulates border cell cluster stiffness as indicated by active Myosin levels and on-time border cell migration (*Aranjuez et al., 2016*). Since Fascin is required in both the nurse cells and the border cells to promote on-time border cell migration (*Lamb et al., 2020*), we wanted to determine which cells Fascin acts in to regulate Myosin activation. To test this, we used the UAS/GAL4 system to express a Fascin RNAi construct to knockdown Fascin in specific cell types (*Figure 6A*) – the germline (*matα GAL4*), somatic (*c355 GAL4*), or border cells (*c306 GAL4*) – and analyzed how loss of Fascin in these different cells affects Myosin activation throughout the follicle. We have previously validated the use of

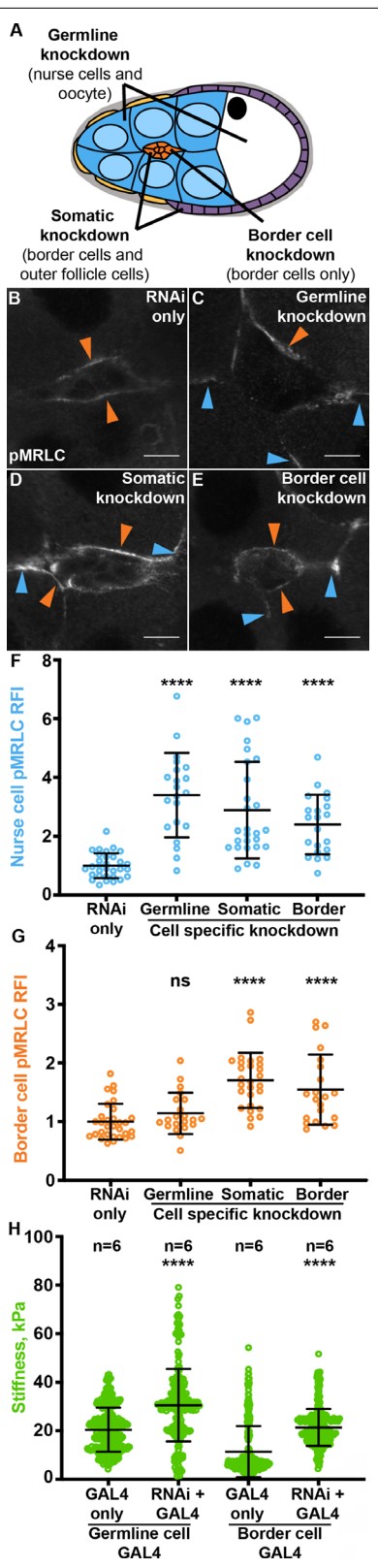

**Figure 6.** Germline Fascin knockdown increases Myosin activation on the nurse cells while somatic Fascin knockdown increases Myosin activation on both the border and nurse cells. (**A**) Schematic of

*Figure 6 continued on next page*

*Figure 6 continued*

cell-specific Fascin knockdown for each GAL4 driver: germline knockdown will knockdown Fascin in the nurse cells (blue) and oocyte (white), somatic cell knockdown will knockdown Fascin in the border cells (orange) and follicle cells (purple and gold), and the border cell knockdown will knockdown Fascin in only the border cells (orange). (**B–E**) Maximum projections of 2–4 confocal slices of Stage 9 follicles of the indicated genotypes stained for phospho-MRLC (pMRLC, white). Orange arrows = pMRLC enrichment on border cell cluster. Blue arrows = pMRLC enrichment on surrounding nurse cells. Scale bars = 10 μm. (**B**) RNAi only (*fascin RNAi/+*). (**C**) Germline knockdown of Fascin (*matα GAL4(3)/fascin RNAi*). (**D**) Somatic cell knockdown of Fascin (*c355 GAL4/+; fascin RNAi/+*). (**E**) Border cell knockdown of Fascin (*c306 GAL4/+; fascin RNAi/+*). (**F, G**) Graphs of quantification of pMRLC intensity at the nurse cell membranes (**F**) and border cell cluster (**G**) in the indicated genotypes. Each circle represents a follicle. Error bars = SD. ns indicates p > 0.05, ****p < 0.0001 (One-way ANOVA with Tukey's multiple comparison test). In **F**, peak pMRLC intensity was quantified at the nurse cell membranes and normalized to phalloidin staining in the same follicle, three measurements were taken per follicle and averaged. In **G**, pMRLC intensity on the border cell cluster was quantified and normalized to background staining in the same follicle. (**H**) Graph of nurse cell stiffness (kPa) of the indicated genotypes as measured by AFM. Each circle represents a single indentation. ****p < 0.0001 (unpaired t-test). Error bars = SD. Fascin regulates Myosin activation in the germline (**C, F, G**) and somatic cells (**D, E, F, G**). Knockdown of Fascin in the germline cells increases Myosin activity and stiffness of the nurse cells (**C, F, H**). Knockdown of Fascin in either all somatic cells or only the border cells increases Myosin activity and stiffness of the nurse cells (**D, E, F, H**), and Myosin activity in the border cell cluster (**D, E, G**). All genotypes are listed in Table 1.

The online version of this article includes the following source data for figure 6:

**Source data 1.** Source data for *Figure 6F-H*.

UAS/GAL4 system to knockdown Fascin in these cell types (*Lamb et al., 2020*).

Based on the literature, we hypothesized knockdown of Fascin in the germline would increase Myosin activation in both the nurse cells and border cells, while knockdown of Fascin in the border cells would only increase Myosin activation in the border cells. We observe, as expected, knockdown of Fascin in the germline results in a significant increase in pMRLC (active MRLC) enrichment on the nurse cell membranes (*Figure 6C*, blue arrows and F, p < 0.0001). However, knockdown of Fascin in the germline unexpectedly fails to alter active MRLC enrichment on the border cell cluster (*Figure 6C*, orange arrow and G, p > 0.05). We next knocked down Fascin in all the somatic cells or just the border cells and anticipated that this would lead to a significant increase in active MRLC on the border cell cluster but not the nurse cells. As expected, we observe a significant increase of active MRLC on the border cell cluster when Fascin is knocked down in the border cells (*Figure 6D, E and G*, orange arrows, p < 0.0001). Surprisingly, knockdown of Fascin in the somatic or just the border cells also significantly increased active MRLC enrichment on the nurse cells (*Figure 6D, E and F*, blue arrows, p < 0.0001). These data surprisingly suggest that knockdown of Fascin in the border cells increases border cell stiffness and this, in turn, induces the stiffening of their substrate, the nurse cells.

Further, we used AFM to directly assess the changes in nurse cell stiffness of our cell specific Fascin knockdowns. Germline knockdown of Fascin results in nurse cells that are 1.5 X stiffer than their GAL4 control (*Figure 6H*, p < 0.0001), while border cell knockdown results in nurse cells that are 1.8 X stiffer than their GAL4 control (*Figure 6H*, p < 0.0001). Together these data demonstrate the unexpected finding that Fascin acts the border cell cluster to regulate the stiffness of the surrounding nurse cell substrate (Figure 8).

## Border cell stiffness controls Myosin activity in its substrate

Our RNAi experiments indicate that Fascin acts primarily in the border cells to control Myosin activation and nurse cell stiffness. If this is true, then restoring Fascin expression in only the somatic cells of a *fascin* mutant follicle, including the border cells, should restore normal Myosin activation in both the border cells and the nurse cells, and normal nurse cell stiffness. Indeed, we find that expressing GFP-Fascin in the somatic cells of *fascin*-null follicles significantly reduces active MRLC enrichment on both the nurse cell and border cell membranes compared to the *fascin*-null control (*Figure 7A–E*, p < 0.0001). Further, restoring Fascin expression in the somatic cells of *fascin*-null follicles significantly reduced the stiffness of the nurse cells compared to the *fascin*-null control (*Figure 7*, 19.4 kPa compared to 38.9 kPa, p < 0.0001). Together our data indicate that Fascin acts in the border cells to regulate the stiffness of both the border cell cluster and its substrate, the nurse cells.

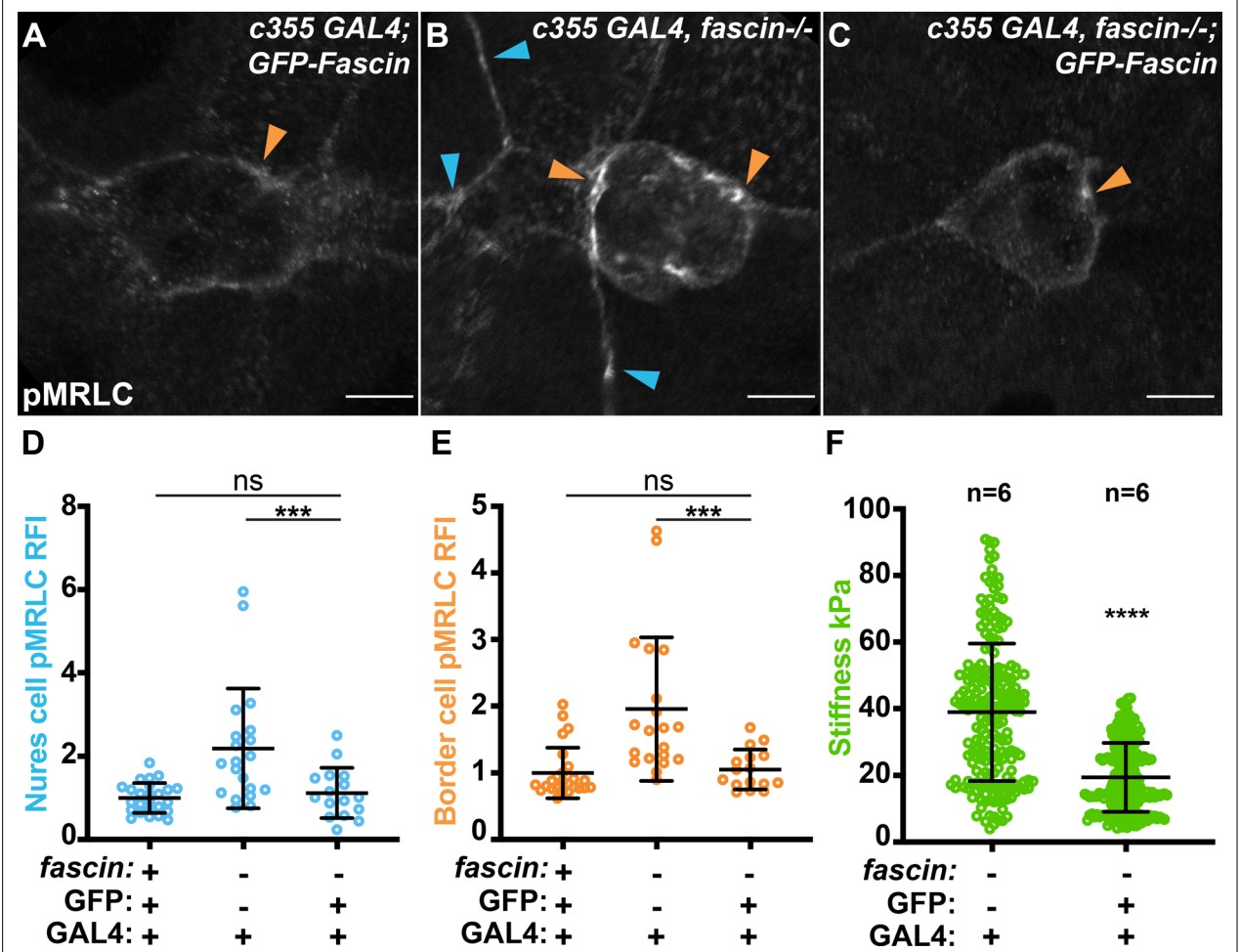

**Figure 7.** Somatic rescue of Fascin reduces nurse cell Myosin activity and stiffness. (**A–C**) Maximum projections of 2–4 confocal slices of Stage 9 follicles of the indicated genotypes stained for phospho-MRLC (pMRLC, white). Blue arrows = pMRLC enrichment on surrounding nurse cells. Orange arrows = pMRLC enrichment on border cell cluster. Scale bars = 10 μm. (**A**) Somatic GFP-Fascin expression (*c355 GAL4/+; UAS-GFP-Fascin/+*). (**B**) *fascin* mutant with somatic GAL4 (*c355 GAL4, fascin^{sn28/sn28}*). (**C**) Somatic GFP-Fascin expression in *fascin* mutant (*c355 GAL4, fascin^{sn28/sn28}; UAS-GFP-Fascin/+*). (**D, E**) Graphs of quantification of pMRLC intensity at the nurse cell membranes (**D**) and border cell cluster (**E**) in the indicated genotypes. Each circle represents a follicle. Error bars = SD. ns indicates $p > 0.05$, ***$p < 0.0001$ (one-way ANOVA with Tukey's multiple comparison test). In **D**, peak pMRLC intensity was quantified at the nurse cell membranes and normalized to phalloidin staining in the same follicle, three measurements were taken per follicle and averaged. In **E**, pMRLC intensity on the border cell cluster was quantified and normalized to background staining in the same follicle. (**F**) Graph of nurse cell stiffness (kPa) of the indicated genotypes as measured by AFM. Each circle represents a single indentation. Error bars = SD. ****$p < 0.0001$ (unpaired t-test). Restoring Fascin expression in the somatic cells of a *fascin* mutant follicle (**C**) significantly reduces activated Myosin enrichment on the nurse cell membranes (**D**) and border cell cluster (**E**) and reduces nurse cell stiffness by AFM (**F**) compared to the *fascin*-null control. All genotypes are listed in Table 1.

The online version of this article includes the following source data and figure supplement(s) for figure 7:

**Source data 1.** Source data for *Figure 7D, E*.

**Figure supplement 1.** Increasing border cell stiffness through activated Rok increases activated Myosin on the nurse cells.

**Figure supplement 1—source data 1.** Source data *Figure 7—figure supplement 1C, D*.

Given the surprising nature of our findings, we next wanted to determine if the border cell regulation of nurse cell stiffness is specific to Fascin or if it is a general principle. To test this idea, we expressed a constitutively active form of Rok (Rok-CAT) in the border cells (*c306* GAL4). Rok is one of the kinases that phosphorylates MRLC to activate Myosin. Thus, expressing constitutively active Rok will increase activation of Myosin, which, in turn, will increase cortical tension and therefore the stiffness of the border cells. We find that expression of constitutively active Rok in the border cells significantly increases active MRLC enrichment on both the nurse cell membranes (***Figure 7—figure supplement***

*1B* compared to 1 A, blue arrows, and C, p < 0.0001) and the border cell cluster (*Figure 7—figure supplement 1B* compared to 1 A, orange arrows and D, p < 0.001). These data suggest that the nurse cells, in general, respond to changes in stiffness of the border cells by altering their own cellular stiffness (*Figure 8*). This non-autonomous regulation of substrate stiffness by the migratory border cells is an unexpected finding.

## Discussion

Using *Drosophila* border cell migration as a model, we provide the first evidence that Fascin limits Myosin activity in vivo to control tissue stiffness *Figure 8*. We find that loss of Fascin significantly increases activated Myosin, and this increase contributes to the border cell migration delays observed in *fascin* mutant follicles during S9. Our data suggest that Fascin's bundling activity is required to limit Myosin activation, supporting the prior proposed model that Fascin tightly bundles F-actin and precludes Myosin from binding to actin filaments (*Elkhatib et al., 2014*). The increased Myosin activity in *fascin* mutants results in substrate stiffening. Using cell-specific knockdown and rescue experiments, we made the suprising finding that Fascin activity in the border cells is necessary and sufficient to regulate Myosin activity and stiffness of the nurse cells. Thus, Fascin activity within the border cells plays a critical role in controlling the balance of forces between the border cells and their substrate, the nurse cells. We also show that this force balance is not specific to Fascin, as directly altering Myosin activity within the border cells phenocopies knockdown of Fascin in these cells. Together our data uncover that in vivo, collectively migrating cells modulate the stiffness of their substrate to control their own migration (*Figure 8*).

Multiple lines of evidence support the model that Fascin is a critical regulator of cellular and tissue stiffness. Specifically, loss of Fascin results in increased active pMRLC on both the border cell and nurse cell membranes, altered MRLC-GFP dynamics on the border cell cluster, and increased nurse cell stiffness as measured by AFM (*Figures 2 and 3*). Interestingly, pMRLC staining and MRLC-GFP time-lapse imaging exhibit distinct patterns (*Figure 2*). In both wild-type and *fascin* mutant follicles, the MRLC-GFP regions are shorter in length than pMRLC puncta. We suspect this difference is due to the MRLC-GFP time-lapse imaging capturing a small period of just activated Myosin, whereas fixation and pMRLC staining captures a longer period of activity. For instance, when Fascin is lost the MRLC-GFP puncta have a longer lifetime, this longer period of Myosin activity may be captured as an increased number of shorter pMRLC puncta. The short and numerous pMRLC puncta may also suggest that Myosin activity in the border cells is not appropriately spatially regulated in *fascin* mutants. These data, in conjunction with our cell-specific RNAi and rescue analyses (*Figures 6 and 7*), reveal that Fascin acts primarily within the border cells to control the stiffness of their substrate, the nurse cells.

Fascin-dependent inhibition of nurse cell Myosin activity and stiffness is essential for on-time border cell migration (*Figure 4*), raising the question of how Fascin regulates Myosin. Our data supports the previously proposed model that Fascin bundled F-actin prevents Myosin binding to F-actin and thereby, restricts Myosin activity (*Elkhatib et al., 2014*). Specifically, we find that expression of the phosphomimetic form of Fascin (S52E), which is unable to bundle F-actin, in *fascin* mutants fails to both inhibit Myosin activation (*Figure 5*) or fully restore migration (*Figure 5—figure supplement 1*). It is important to note that phosphorylated Fascin, and likely phosphomimetic Fascin, also exhibits reduced F-actin binding by in vitro assays (*Yamakita et al., 1996*), raising the possibility that Fascin binding to F-actin, without bundling filaments, is sufficient to inhibit Myosin activation. Further experiments are needed to fully elucidate how Fascin limits Myosin activity.

Our discovery that Fascin limits Myosin activity in vivo is unlikely to be restricted to *Drosophila*. Indeed, both Fascin and Myosin play critical roles during cancer metastasis (*Hashimoto et al., 2011*; *Aguilar-Cuenca et al., 2014*; *Ma and Machesky, 2015*). Increased Myosin activation and consequently, increased stiffness are a common phenotype observed in cancer cells and their substrate (*Tse et al., 2012*; *Aguilar-Cuenca et al., 2014*; *van Helvert and Friedl, 2016*; *Ren et al., 2021*). Increased substrate stiffness promotes migration in a wide range of cancers, suggesting increased Myosin activity can lead to increased cancer metastasis (*Aguilar-Cuenca et al., 2014*; *Emon et al., 2018*; *Mierke, 2020*; *Ren et al., 2021*). Additionally, Fascin is highly expressed in many types of cancers, notably carcinomas (*Hashimoto et al., 2011*; *Ma and Machesky, 2015*). High Fascin expression in these cancers is correlated with increased migration (*Grothey et al., 2000*; *Hashimoto et al., 2007*), invasion (*Adams et al., 1999*; *Minn et al., 2005*), and metastasis (*Li et al., 2014*; *Alburquerque-González*

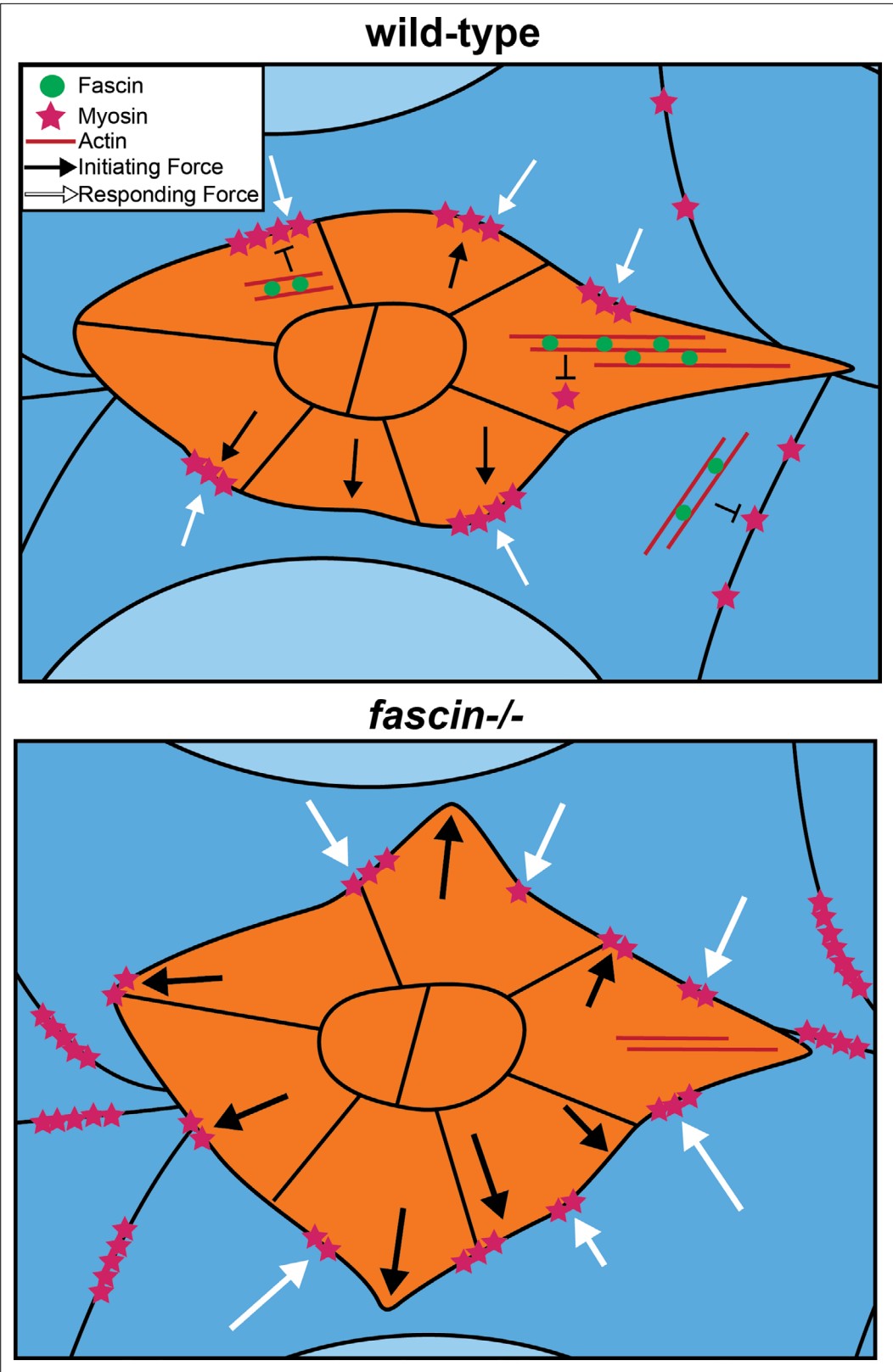

**Figure 8.** Proposed model for Fascin limiting Myosin activity to control substrate stiffness during border cell migration. In wild-type border cell clusters (orange), Fascin (green circles) bundles F-actin (red lines) to limit Myosin activity (magenta stars) on the border cell cluster and on the nurse cell membranes. Myosin activity in the border cell cluster generates forces (black arrows) that pushes on the nurse cells which results in the nurse cells

*Figure 8 continued on next page*

*Figure 8 continued*

responding with force (white arrows). This balance of forces is required for on-time migration. In *fascin* mutant follicles, Myosin activity on the border cell cluster is increased, driving increased Myosin activity on the nurse cells. This imbalance of forces between the border cell cluster and the nurse cell substrate impairs border cell migration.

*et al., 2020*). However, according to our model, increased Fascin would reduce Myosin activity. Our finding that Fascin-dependent bundling is required to limit Myosin activity and substrate stiffness suggests that phosphorylated Fascin may promote cancer metastasis by allowing high Myosin activation and potentially other bundling-independent functions. Supporting this idea, expression of a S39 phosphomimetic form of Fascin, which cannot bundle F-actin, promotes human colon carcinoma migration (*Hashimoto et al., 2007*), suggesting phosphorylated Fascin could promote cancer metastasis by allowing increased Myosin activation and cell stiffness.

Our finding that phosphomimetic Fascin only partially rescues the migration delay in *fascin* mutants suggests that non-bundling roles of Fascin also contribute to border cell migration. Indeed, Fascin has many functions besides F-actin bundling, such regulating microtubules (*Villari et al., 2015*) and acting within the nucleus (*Groen et al., 2015*). Additionally, S52 phosphorylated Fascin functions as an adaptor for the Linker of the Nucleoskeleton and Cytoskeleton (LINC) Complex (*Groen et al., 2015*; *Jayo et al., 2016*). This LINC Complex role of Fascin is required for nuclear shape changes necessary for mammalian single-cell invasive migration (*Jayo et al., 2016*), raising the idea that Fascin may be similarly required for the invasion of the border cells between the nurse cells. Further experiments are needed to understand how the different functions of Fascin are coordinated to promote migration.

Our results suggest that increased stiffness in the border cell cluster affects the stiffness of its substrate, the nurse cells (*Figures 6 and 7*). This non-autonomous function of the border cells in altering the stiffness of the nurse cells was unexpected, as previous data suggested the nurse cells exert force on the border cells and the border cells respond to this force (*Aranjuez et al., 2016*). This balance of forces is necessary to promote the migration of the cluster through the tightly packed nurse cells (*Aranjuez et al., 2016*). Specifically, a previous study observed that overexpression of a Rho GEF in the nurse cells, which both increased Myosin activation and caused the nurse cells to change their shape and become more circular, ultimately impairs border cell migration (*Aranjuez et al., 2016*). As we do not observe any obvious changes in nurse cell shape when Fascin is lost or knocked down in the nurse cells, it may be that loss of Fascin does not cause a severe enough change in nurse cell Myosin activity and cell stiffness to cause the border cell cluster to respond. Instead, our data suggest that the border cells play a larger role in this balance of forces by exerting force on the nurse cells to control nurse cell stiffness. This interaction could potentially allow the border cell cluster to stiffen the nurse cells as the cluster migrates. Interestingly, in the context of cancer cell migration, a stiffer substrate often promotes cell migration (*Parekh and Weaver, 2016*; *Oakes, 2018*; *Ren et al., 2021*). Further, there is growing evidence that one means of directing migration is a gradient of substrate stiffness, such that cells move from softer to stiffer substrates; this is termed durotaxis (*Sunyer and Trepat, 2020*; *Shellard and Mayor, 2021*). Indeed, durotaxis has emerged as a property of collectively migrating cells. Specifically, it has been suggested that clusters of migrating cells are better able to sense differences in stiffness and respond more effectively (*Martinez et al., 2016*; *Sunyer et al., 2016*). Therefore, it is tempting to speculate that the border cells exert force on the nurse cells to stiffen them to aid in migration.

A key remaining question is whether border cells regulates substrate stiffness in a spatial and/or temporal manner. Specifically, do the border cells push on the nurse cells as they are migrating, driving a wave of local Myosin activation and stiffening of the nurse cell substrate? In other systems, such local stiffening appears to be cell-type and context specific, and can occur at the front or back of the migrating cells (*Doyle et al., 2021*). Based on both our fixed- and live-imaging of Myosin activity, we believe the stiffening of the nurse cell substrate occurs on all sides of the border cell cluster. This phenotype is consistent in wild-type and *fascin*-null follicles with *fascin*-null follicles displaying an overall increase in pMRLC. However, our current means of assessing Myosin activity and cellular stiffness lack the resolution necessary to determine how Myosin activation and changes in stiffness propagate through the nurse cells. Future studies using tools that allow Myosin activity in the border cells to be distinguished from that in its nurse cell substrate, in conjunction with higher resolution, rapid, non-photobleaching microscopy approaches, and cell-specific means of assessing individual

nurse cell stiffness, such as laser ablation recoil velocity assessments, are need to uncover the detailed dynamics of how the balance of forces between the border cells and their substrate, the nurse cells, drives collective cell migration.

It remains unclear how Myosin activity at the cell cortex is connected to cellular adhesions in the border cells. Prior work indicates that the border cells migrate directly on the nurse cells, as there is little to no ECM present (*Medioni and Noselli, 2005*), and thus, Integrin-based ECM adhesions are not critical for border cell migration (*Dinkins et al., 2008*; *Llense and Martín-Blanco, 2008*; *Montell et al., 2012*). Instead, border cells may utilize E-Cadherin based adhesions (*Niewiadomska et al., 1999*; *Cai et al., 2014*) or perhaps uncharacterized means to drive migration. Thus, additional work is needed to determine the cellular connections mediating Myosin-dependent force transmission between the border cells and the nurse cells.

While it is clear that the balance of forces between the border cells and the nurse cells is critical for border cell migration, the mechanisms by which force imbalances impair migration remain poorly understood. We speculate that the increased Myosin activity in *fascin* mutants delays migration by impacting delamination and protrusion dynamics. We previously found that Fascin is required for on-time delamination of the border cell cluster from the follicular epithelium and for restricting the number and location of protrusions to the leading edge of the border cell cluster (*Lamb et al., 2020*). Similarly, both loss and constitutive activation of Myosin within the border cells delays delamination and causes excessive and misdirected protrusions (*Majumder et al., 2012*; *Aranjuez et al., 2016*; *Mishra et al., 2019*). These data suggest that it is not only the level of Myosin activity but is ability to cycle between active and inactive states that contributes to these two aspects of border cell migration. Based on our data, we suspect altering Myosin activity in the border cells ultimately changes the stiffness of the nurse cells. Too little activation of Myosin would result in a soft substrate and too much would result in a stiff substrate. Such changes in substrate stiffness could alter the polarization of the cluster, resulting in mislocalized and increased protrusions which not only delay migration but impair delamination. Supporting this idea, Myosin regulates active Rac polarization within the border cells (*Mishra et al., 2019*). Rac activation is highest in the leading cell of the border cell cluster and is require to generate forward directed protrusions (*Fulga and Rørth, 2002*; *Bianco et al., 2007*; *Mishra et al., 2019*). Increased Myosin activation in the border cell cluster disrupts this polarization, resulting in mislocalized protrusions (*Mishra et al., 2019*). This loss of polarization could function cell-autonomously, but, based on our data, it may also increase nurse cell stiffness. Such increased substrate stiffness could impair delamination and cause mislocalized protrusions by physically altering the topography of the nurse cells, which has recently been shown to be critical for border cell migration and forward directed protrusions (*Dai et al., 2020*). Additionally, increased substrate stiffness could disrupt durotactic signaling or alter the diffusion of the ligands directing migration. Thus, Fascin's role in limiting Myosin activation likely contributes to the delayed delamination and aberrant mislocalized protrusions observed during border cell migration in *fascin*-null follicles.

The mechanical communication between migrating cells and their substrate is a growing area of research. The overarching premise in the field has been that substrate stiffness regulates the mechanical properties of the migrating cells and thereby, alters their ability to migrate. For example, in a model of breast cancer cell migration, high substrate stiffness promotes migration (*Ren et al., 2021*). Additionally during zebrafish development, the underlying mesoderm must stiffen to induce the epithelial to mesenchymal transition (EMT) and migration of the neural crest cells (*Barriga et al., 2018*). Together these studies highlight the current paradigm that substrate stiffness is the driving force that regulates the migrating cells to control their migration. However, the roles of the migrating cells in controlling their substrate stiffness are less understood. Numerous studies support that migrating cells can degrade surrounding ECM, creating paths for easier migration (for example: *Wolf et al., 2007*); such changes likely decrease substrate stiffness in the local environment. Migrating cells also pull on their local environment, applying a strain on the environment and aligning ECM fibers, which ultimately causes a local increase in substrate stiffness (*Hall et al., 2016*; *van Helvert and Friedl, 2016*). Notably, collectively migrating cells exert 4 x more force on their environment than single cells (*van Helvert and Friedl, 2016*). These studies indicate migrating cells can, at least locally, influence the stiffness of their substrate. This increase in substrate stiffness promotes cell migration, increasing migratory cell force generation in a process termed mechanoreciprocity (*van Helvert et al., 2018*). Whether migrating cells control substrate and underlying tissue stiffness in native, physiological contexts, and whether local stiffness changes are

propagated through the tissue remain poorly understood. Supporting that this may occur, cancer cells drive stromal changes, including increased fibrosis which stiffens the tissue (*van Helvert et al., 2018*; *Chandler et al., 2019*; *Piersma et al., 2020*). These changes have been proposed as potential mechanism by which cancer cells drive their own invasion and make environments supportive of metastatic colonization (*Cox and Erler, 2014*). Our finding that Fascin activity in the migrating border cells controls substrate stiffness to promote migration positions *Drosophila* border cell migration as a robust system to uncover the mechanisms controlling this means of force balance.

Here, we propose that migrating cells modulate their own stiffness to regulate substrate stiffness. Our findings suggest that during collective cell migrations, such as those during development and cancer metastasis, the migrating cells apply force to induce the stiffening of their substrate, this results in a reciprocal mechanical communication between the migrating cells and their substrate which drives migration. Further, we demonstrate that Fascin, an F-actin bundling protein, limits the activity of Myosin in the migrating cells to regulate substrate stiffness. Overall, our findings expand our understanding of the mechanical relationship between migrating cells and their substrate, shifting the paradigm in the field from the substrate controlling migrating cell stiffness and thereby, migration, to the migrating cells playing a key role in altering their environment and substrate stiffness to promote their own migration.

# Materials and methods

**Key resources table**

| Reagent type (species) or resource | Designation | Source or reference | Identifiers | Additional information |
|---|---|---|---|---|
| Genetic reagent (*Drosophila melanogaster*) | $y^1w^1$ | Bloomington *Drosophila* Stock Center | BDSC Cat # 1495 RRID:BDSC_1495 | |
| Genetic reagent (*D. melanogaster*) | *fascin^{sn28/sn28}* | other | FBgn0003447 | from J. Zanet |
| Genetic reagent (*D. melanogaster*) | *fTRG sqh* | Vienna *Drosophila* Resource Center | VRDC Cat # 318,484 | fTRG 10075 |
| Genetic reagent (*D. melanogaster*) | *oskar GAL4 (2)* | Bloomington *Drosophila* Stock Center | BDSC Cat # 44,241 RRID:BDSC_44241 | Anne Ephrussi |
| Genetic reagent (*D. melanogaster*) | *UAS-sqh RNAi* | Bloomington *Drosophila* Stock Center | BDSC Cat # 33,892 RRID:BDSC_33892 | |
| Genetic reagent (*D. melanogaster*) | *actin 5 c GAL4* | Bloomington *Drosophila* Stock Center | BDSC Cat # 8,807 RRID:BDSC_8807 | |
| Genetic reagent (*D. melanogaster*) | *UAS-GFP-Fascin* | *Zanet et al., 2009* PMID:19592575 | | from J. Zanet |
| Genetic reagent (*D. melanogaster*) | *UAS-GFP-Fascin-S52E* | *Zanet et al., 2009* PMID:19592575 | | from J. Zanet |
| Genetic reagent (*D. melanogaster*) | *UAS-Fascin-RNAi* | Bloomington *Drosophila* Stock Center | BDSC Cat # 42,615 RRID:BDSC_42615 | |
| Genetic reagent (*D. melanogaster*) | *matα GAL4* | Bloomington *Drosophila* Stock Center | BDSC Cat # 7,063 RRID:BDSC_7063 | |
| Genetic reagent (*D. melanogaster*) | *c355 GAL4* | Bloomington *Drosophila* Stock Center | BDSC Cat # 3,750 RRID:BDSC_3750 | |
| Genetic reagent (*D. melanogaster*) | *c306 GAL4* | Bloomington *Drosophila* Stock Center | BDSC Cat # 3,743 RRID:BDSC_3743 | |
| Genetic reagent (*D. melanogaster*) | *UAS-Rok-CAT* | Bloomington *Drosophila* Stock Center | BDSC Cat # 6,669 RRID:BDSC_6669 | |
| Antibody | rabbit polyclonal anti-Phospho-Myosin Light Chain 2 (Ser19) | Cell Signaling | #3,671 RRID:AB_330248 | (1:100) |
| Antibody | mouse monoclonal anti-Hu li tai shao (Hts) | Developmental Studies Hybridoma Bank | 1B1 RRID:AB_528070 | (1:50) |

*Continued on next page*

*Continued*

| Reagent type (species) or resource | Designation | Source or reference | Identifiers | Additional information |
|---|---|---|---|---|
| Antibody | mouse monoclonal anti-Fasciclin III | Developmental Studies Hybridoma Bank | 7G10 RRID:AB_528238 | (1:50) |
| Antibody | rat monoclonal anti-Vasa | Developmental Studies Hybridoma Bank | RRID:AB_760351 | (1:100) |
| Antibody | rabbit polyclonal anti-Zipper | *Wheatley et al., 1995* PMID: 7601006 | | (1:10000) |
| Antibody | rabbit polyclonal anti-GFP | Torrey Pines Biolabs, Inc | TP401 RRID:AB_10013661 | (1:2000) |
| Antibody | goat polyclonal anti-GFP | Fitzgerald Industries International | 70R-GG001 RRID:AB_1286216 | (1:2000) |
| Antibody | goat polyclonal Alexa Fluor 488 anti-mouse | Thermo Fischer Scientific | A-11001 RRID:AB_2534069 | (1:500) |
| Antibody | goat polyclonal Alexa Fluor 568 anti-mouse | Thermo Fischer Scientific | A-11004 RRID:AB_2534072 | (1:500) |
| Antibody | goat polyclonal Alexa Fluor 488 anti-rabbit | Thermo Fischer Scientific | A-11034 RRID:AB_2576217 | (1:500) |
| Antibody | goat polyclonal Alexa Fluor 568 anti-rabbit | Thermo Fischer Scientific | A-11036 RRID:AB_10563566 | (1:500) |
| Antibody | donkey polyclonal Alexa Fluor 488 anti-goat | Thermo Fischer Scientific | A-11055 RRID:AB_2534102 | (1:500) |
| Antibody | Peroxidase-AffiniPure goat polyclonal anti-rabbit | Jackson ImmunoResearch Laboratories | 111-035-003 RRID:AB_2313567 | (1:5000) |
| Antibody | Peroxidase-AffiniPure goat polyclonal anti-rat | Jackson ImmunoResearch Laboratories | 112-035-003 RRID:AB_2338128 | (1:5000) |
| Chemical compound, drug | 4′,6-Diamidino-2-phenylindole (DAPI) | Millipore Sigma | D9542 | 5 mg/ml |
| Chemical compound, drug | Alexa Flour 568 or 647 Phalloidin | Thermo Fischer Scientific | A12380 or A22287 | (1:250 or 1:500) |
| Chemical compound, drug | Y-27632 | Millipore Sigma | Y0503 | 200 µM |
| Software, algorithm | FIJI | *Schindelin et al., 2012* PMID:22743772 | RRID:SCR_002285 | |
| Software, algorithm | Prism 8 and 9 | https://www.graphpad.com/ | RRID:SCR_002798 | |
| Software, algorithm | Adobe Photoshop CC | https://ww.adobe.com/ | RRID:SCR_014199 | |
| Software, algorithm | Adobe Illustrator CC 25.2.3 | https://ww.adobe.com/ | RRID:SCR_010279 | |
| Software, algorithm | LAS AS SPE Core | Leica | | |
| Software, algorithm | ZEN Axio Observer.Z1 | Zeiss | | |

## Fly stocks

Fly stocks were maintained on cornmeal/agar/yeast food at 21 °C, except where noted. Before immunofluorescence staining and live imaging, flies were fed wet yeast paste daily for 2–4 days. Unless otherwise noted, *yw* was used as the wild-type control. The following stocks were obtained from the Bloomington Stock Center (Bloomington, IN): *matα* GAL4 (third chromosome), *c355* GAL4, *c306* GAL4, *actin5C* GAL4, *UASp-RNAi-Fascin* (TRiP.HMS02450), *UASp-Sqh-RNAi* (TRiP.HMS00437), and *UASp-Rok-CAT*. The *fTRG sqh* stock was obtained from the Vienna *Drosophila* Resource Center. The *fascin^{sn28}* line was a generous gift form Jennifer Zanet (Université de Toulouse, Toulouse, France *Zanet et al., 2012*), the *oskar* GAL4 line (second chromosome) was a generous gift from Anne Ephrussi (European Molecular Biology Laboratory, Heidelber, Germany *Telley et al., 2012*), and the *UASp-GFP-Fascin* and *UASp-GFP-Fascin-S52E* lines were a generous gift from Francois Payre (Université de Toulouse, Toulouse, France *Zanet et al., 2009*). For germline expression during S9, either *matα* GAL4 or *oskar* GAL4 were utilized interchangeably. Expression of *UASp-RNAi-Fascin* was achieved

by crossing to *matα* GAL4, *c355* GAL4, and *c306* GAL4, maintaining crosses at 25 °C and progeny at 29 °C for 3 days. Expression of *UASp-Sqh-RNAi* was achieved by crossing to *oskar* GAL4, maintaining crosses at 25 °C and progeny at 29 °C for 3 days. The *sn28, c355* GAL4 flies were generated previously (*Lamb et al., 2020*). Expression of *UASp-GFP-Fascin* or *UASp-GFP-Fascin-S52E* was achieved by crossing to *actin5C* GAL4, crosses were maintained at 25 °C and progeny at 29 °C for 2 days. The specific genotypes for each experiment are listed in *Table 1*.

## Immunofluorescence

Whole-mount *Drosophila* ovary samples (approximately five flies per experiment) were dissected into Grace's insect media (Lonza, Walkersville, MD) and fixed for 10 min at room temperature in 4 % paraformaldehyde in Grace's insect media. Briefly, samples were blocked using antibody wash (1 X phosphate-buffered saline, 0.1 % Triton X-100, and 0.1 % bovine serum albumin) six times for 10 min each. Primary antibodies were diluted with antibody wash and incubated overnight at 4 °C. The following primary antibodies were obtained from the Developmental Studies Hybridoma Bank (DSHB) developed under the auspices of the National Institute of Child Health and Human Development and maintained by the Department of Biology, University of Iowa (Iowa City, IA): mouse anti-Hts 1:50 (1B1, Lipshitz, HD *Zaccai and Lipshitz, 1996*), mouse anti-FasIII 1:50 (7G10, Goodman, C *Patel et al., 1987*); mouse anti-Fascin 1:20 (sn7c, Cooley, L *Cant et al., 1994*). Additionally, the following primary antibody was used: rabbit anti-GFP 1:2000 (pre-absorbed on *yw* ovaries at 1:20 and used at 1:100; Torrey Pines Biolabs, Inc, Secaucus, NJ). After six washes in Triton antibody wash (10 min each), secondary antibodies were incubated overnight at 4 °C or for ~4 hr at room temperature. The following secondary antibodies were used at 1:500: AlexaFluor (AF)488::goat anti-mouse, AF568::goat anti-mouse, AF488::goat anti-rabbit, AF568::goat anti-rabbit (Thermo Fischer Scientific). AF647-, or AF568-conjugated phalloidin (Thermo Fischer Scientific) was included with primary and secondary antibodies at a concentration of 1:250. After six washes in antibody wash (10 minutes each), 4',6-diamidino-2-phenylindole (DAPI, 5 mg/ml) staining was performed at a concentration of 1:5,000 in 1 X PBS for 10 minutes at room temperature. Ovaries were mounted in 1 mg/ml phenylenediamine in 50 % glycerol, pH 9 (*Platt and Michael, 1983*). All experiments were performed a minimum of three independent times.

Active-MRLC staining was performed using a modified protocol provided by Jocelyn McDonald (*Majumder et al., 2012*; *Aranjuez et al., 2016*). Briefly, ovaries were fixed for 20 min at room temperature in 8 % paraformaldehyde in 1 X phosphate-buffered saline (PBS) and 0.5 % Triton X-100. Samples were blocked by incubating in Triton antibody wash (1XPBS, 0.5 % Triton X-100, and 5 % bovine serum albumin) for 30 min. Primary antibodies were incubated for 48 hr at 4 °C. The rabbit anti-pMRLC (S19; Cell Signaling, Davers, MA) was diluted 1:100 in Triton antibody wash. Anti-Fascin (sn7c, 1:20) was sometimes added to the primary antibody solution to differentiate between wild-type and *fascin*-null follicles in the same sample or to confirm Fascin RNAi knockdown. In other cases, anti-Hts (1B1, 1:50) and anti-FasIII (7G10, 1:50) were added to the primary antibody solution to allow for visualization of the border cell cluster boundaries. After six washes in Triton antibody wash (10 min each), the secondary antibodies were diluted 1:500 in Triton antibody wash and incubated overnight at 4 °C. Alexa Fluor 647–phalloidin (Invitrogen, Life Technologies, Grand Island, NY) was included with both primary and secondary antibodies at a concentration of 1:250; this allowed for visualization of the border cell cluster boundaries. Samples were washed six times in Triton antibody wash (10 min each) and the stained with DAPI and mounted as described above.

## Image acquisition and processing

Microscope images of fixed *Drosophila* follicles were obtained using LAS AS SPE Core software on a Leica TCS SPE mounted on a Leica DM2500 using an ACS APO 20 x/0.60 IMM CORR -/D objective (Leica Microsystems, Buffalo Grove, IL) or using Zen software on a Zeiss 700 LSM mounted on an Axio Observer.Z1 using a Plan-Apochromat 20 x/0.8 working distance (WD) = 0.55 M27 or a EC-Plan-Neo-Fluar 40 x/1.3 oil objective (Carl Zeiss Microscopy, Thornwood, NY). Maximum projections (two to four confocal slices), merged images, rotations, and cropping were performed using ImageJ software (*Abramoff et al., 2004*). S9 follicles were identified during fixed imaging by the size of the follicle (~150–250 µm), the position and morphology of the outer follicle cells, and presence of a border cell cluster. The beginning of S10 was defined as when the anterior most outer follicle cells reached the nurse cell-oocyte boundary and flattened.

**Table 1.** Genotype by figures.

List of genotype show in the figures.

| Figure | Panel | Genotype |
|---|---|---|
| | B | *yw* |
| *Figure 1* | C | *fascin$^{sn28/sn28}$* |
| | A-A" | *yw* |
| | B-B" | *fascin$^{sn28/sn28}$* |
| | C-F | *yw* |
| | | *fascin$^{sn28/sn28}$* |
| | G-G'''| *fascin$^{sn28}$/+; +/sqh-GFP* |
| | H-H'''| *fascin$^{sn28/sn28}$; +/sqh-GFP* |
| | | *fascin$^{sn28}$/+; +/sqh-GFP* |
| *Figure 2* | I | *fascin$^{sn28/sn28}$; +/sqh-GFP* |
| | A-B | *fascin$^{sn28/sn28}$* |
| | C-F | *yw* |
| *Figure 2—figure supplement 1* | | *fascin$^{sn28/sn28}$* |
| | C | *yw* |
| | | *yw* |
| *Figure 3* | D | *fascin$^{sn28/sn28}$* |
| | B | *yw* |
| | C-D | *fascin$^{sn28/sn28}$* |
| | E | *fascin$^{sn28/sn28}$; oskar GAL4 (2)/+* |
| | F | *fascin$^{sn28/sn28}$; oskar GAL4 (2)/UAS-sqh-RNAi* |
| | | *yw* |
| | G | *fascin$^{sn28/sn28}$* |
| | | *oskar GAL4 (2)/+* |
| | | *oskar GAL4 (2)/UAS-sqh-RNAi* |
| | | *fascin$^{sn28/sn28}$; oskar GAL4 (2)/+* |
| *Figure 4* | H | *fascin$^{sn28/sn28}$; oskar GAL4 (2)/UAS-sqh-RNAi* |
| *Figure 4—figure supplement 1* | | *yw* |
| | A-B | *fascin$^{sn28/sn28}$* |
| | C-D | *oskar GAL4 (2)/+* |
| | | *oskar GAL4 (2)/UAS-sqh-RNAi* |
| | | *fascin$^{sn28/sn28}$; oskar GAL4 (2)/+* |
| | | *fascin$^{sn28/sn28}$; oskar GAL4 (2)/UAS-sqh-RNAi* |

*Table 1 continued*

| Figure | Panel | Genotype |
|---|---|---|
| | A | *fascin^sn28/sn28; actin 5c GAL4/+* |
| | B | *fascin^sn28/sn28; actin 5c GAL4/UAS-GFP-Fascin* |
| | C | *fascin^sn28/sn28; actin 5c GAL4/UAS-GFP-Fascin-S52E* |
| | | *actin 5c GAL4/+* |
| | | *fascin^sn28/sn28; actin 5c GAL4/+* |
| | | *fascin^sn28/sn28; actin 5c GAL4/UAS-GFP-Fascin* |
| *Figure 5* | D-E | *fascin^sn28/sn28; actin 5c GAL4/UAS-GFP-Fascin-S52E* |
| | A | *fascin^sn28/sn28; actin 5c GAL4/UAS-GFP-Fascin* |
| | B | *fascin^sn28/sn28; actin 5c GAL4/UAS-GFP-Fascin-S52E* |
| | | *actin 5c GAL4/+* |
| | | *fascin^sn28/sn28; actin 5c GAL4/+* |
| | | *fascin^sn28/sn28; actin 5c GAL4/UAS-GFP-Fascin* |
| *Figure 5—figure supplement 1* | C | *fascin^sn28/sn28; actin 5c GAL4/UAS-GFP-Fascin-S52E* |
| | B | *+/UAS-Fascin-RNAi (3)* |
| | C | *matα GAL4 (3)/UAS-Fascin-RNAi (3)* |
| | D | *c355 GAL4/+; +/UAS-Fascin-RNAi (3)* |
| | E | *c306 GAL4/+; +/UAS-Fascin-RNAi (3)* |
| | | *+/UAS-Fascin-RNAi (3)* |
| | | *matα GAL4 (3)/UAS-Fascin-RNAi (3)* |
| | | *c355 GAL4/+; +/UAS-Fascin-RNAi (3)* |
| *Figure 6* | F-H | *c306 GAL4/+; +/UAS-Fascin-RNAi (3)* |
| | A | *c355 GAL4/+; +/UAS-GFP-Fascin* |
| | B | *c355 GAL4, fascin^sn28/sn28* |
| | C | *c355 GAL4, fascin^sn28/sn28; +/UAS-GFP-Fascin* |
| | | *c355 GAL4/+; +/UAS-GFP-Fascin* |
| | | *c355 GAL4, fascin^sn28/sn28* |
| | D-E | *c355 GAL4, fascin^sn28/sn28; +/UAS-GFP-Fascin* |
| | | *c355 GAL4, fascin^sn28/sn28* |
| *Figure 7* | F | *c355 GAL4, fascin^sn28/sn28; +/UAS-GFP-Fascin* |
| | A | *c306 GAL4/+* |
| | B | *c306 GAL4/+; +/UAS-Rok-CAT* |
| | | *c306 GAL4/+* |
| | | *+/UAS-Rok-CAT* |
| *Figure 7—figure supplement 1* | C-D | *c306 GAL4/+; +/UAS-Rok-CAT* |
| *Video 1* | | *fascin^sn28/+; +/sqh-GFP* |
| *Video 2* | | *fascin^sn28/sn28; +/sqh-GFP* |

## Quantification of fixed imaging for border cell migration

Quantification of the migration index of border cell migration was performed as described previously (*Fox et al., 2020*; *Lamb et al., 2020*). Briefly, measurements of S9 follicles were performed on

confocal image stacks of follicles stained with anti-Hts and anti-FasIII or phalloidin. Measurements of migration distances were obtained from maximum projections of 2–4 confocal slices of deidentified 20 x confocal images using ImageJ software (*Abramoff et al., 2004*). Briefly, a line segment was drawn from the anterior end of the follicle to the front or posterior of the border cell cluster and the distance in microns measured, this was defined as the distance of border cell migration. Additionally, a line segment was drawn from the anterior end of the follicle to the anterior end of the main-body follicle cells and the distance measured, this was defined as the distance of the outer follicle cells. Lastly, the entire follicle length was measured along the anterior-posterior axis. The migration index was calculated in Excel (Microsoft, Redmond, WA) by dividing the border cell distance by the follicle cell distance. Cluster length was determined by measuring the distance from the front to the rear of the border cell cluster (detached cells were not included). Data was compiled, graphs generated, and statistical analysis performed using Prism (GraphPad Software).

## pMRLC quantifications

Fluorescence intensity analyses were performed on maximum projections of 3 confocal slices of 40 x confocal images using ImageJ software. Concurrent Fascin or border cell staining (Hts and FasIII) was used to define the boundaries of the border cell cluster. For nurse cell intensity, three line segments per follicle were drawn across nurse cell-nurse cell membranes on maximum projections of 2–3 confocal slices of follicles stained for pMRLC and phalloidin. The fluorescent intensity peak for pMRLC was determined for each line and normalized to phalloidin intensity at the same point. These three values were then averaged for a single image. Averages were then normalized to the wild-type average for each experiment due to experimental variability (for example quantification see *Figure 2—figure supplement 1*). For border cell intensity, the border cell cluster was traced using the phalloidin stain and the mean fluorescence intensity for pMRLC was measured for this shape and this was then normalized to the mean fluorescence intensity of pMRLC of the same shape in the nurse cell cytoplasm (for example quantification see *Figure 2—figure supplement 1*). For the puncta number and length, puncta on the border cell cluster were manually counted and length measured from a maximum projection image using ImageJ software. Data was compiled, graphs generated, and statistical analysis performed using Prism (GraphPad Software).

## Live imaging

Whole ovaries were dissected from flies fed wet yeast paste for 2–3 days and maintained at 25 °C until the last 16–24 hr when they were moved to 29 °C. Genotypes used for live imaging were *sn28/FM7; sqh-GFP* and *sn28/sn28; sqh-GFP*. Ovaries were dissected in Stage 9 (S9) medium (*Prasad and Montell, 2007*): Schneider's medium (Life Technologies), 0.6 x penicillin/streptomycin (Life Technologies), 0.2 mg/ml insulin (Sigma-Aldrich, St. Louis, MO), and 15 % fetal bovine serum (Atlanta Biologicals, Flowery Branch, GA). S9 follicles were hand dissected and embedded in 1.25 % low-melt agarose (IBI Scientific, Peosta, IA) made with S9 media on a coverslip-bottom dish (MatTek, Ashland, MA). Just prior to live imaging, fresh S9 media was added to coverslip-bottom dish. Live imaging was performed with Zen software on a Zeiss 700 LSM mounted on an Axio Observer.Z1 using a Plan-Apochromat 20 x/0.8 working distance (WD) = 0.55 M27 (Carl Zeiss Microscopy, Thornwood, NY) with a 2 x zoom. Images were acquired every 30 seconds for at least 1 hour for *Sqh-GFP* flies. Maximum projections (2–5 confocal slices), merge images, rotations, and cropping were performed using ImageJ software (*Abramoff et al., 2004*) To aid in visualization live imaging videos were brightened by 50 % in Photoshop (Adobe, San Jose, CA).

## Quantification of live imaging

Quantifications of live imaging videos were performed in ImageJ (*Abramoff et al., 2004*) using maximum projection of 2–5 confocal slices from time-lapse videos of border cell migration. For MRLC-GFP live imaging, puncta lifetime was defined by the amount of time elapsed from when a punctum first appeared to when it disappeared completely. Data were compiled, graphs generated, and statistical analysis performed using Prism (GraphPad Software).

## Atomic force microscopy (AFM) nanoindentation on *Drosophila* follicles

Whole ovaries were dissected from flies fed wet yeast paste for 2–3 days. Ovaries were dissected in S9 medium (*Prasad and Montell, 2007*), as described above. S9 follicles were hand isolated and mounted on poly-D-lysine coated 35 mm round glass coverslips. Force spectroscopy data were collected using a molecular force probe 3D (Asylum research) AFM in a liquid cell. AFM force spectroscopy was performed in a buffered solution within 1–2 hr after submersion. A new silicon nitride AFM probe (Bruker, DNP-10) was used for every experiment with a nominal spring constant of 0.12 N/m and a half cone angle of 20 degrees. Actual spring constant was calibrated using the built-in thermal noise method prior to measurement collection in each experiment. S9 follicles were located using the top view video camera and AFM force versus indentation data were collected on the middle of the follicle. The force data were recorded with a 0.6–1.2 µm/s tip approach velocity and a maximum force ranging from 1 to 5 nN. For each genotype, two to three follicles were probed per experiment for three independent experiments; a total of six to nine follicles were probed per genotype. In each region, five to eight different positions with 2–10 µm separations were probed. For each position, 3–10 multiple repeated force curves were recorded. Two stiffness values of follicles were determined by fitting the approach data of two separate tip depth force-indentation curves to the rearranged form of the Hertzian elastic contact model (*Heinrich, 1882*). These two force-indentation ranges were selected to measure the stiffness of the basement membrane (20–100 nm) and underlying nurse cells (310–550 nm) and are similar to previous studies (*Chlasta et al., 2017*; *Crest et al., 2017*; *Chen et al., 2019*). Poisson's ratios of 0.5 and 0.25 were assumed for the follicles and AFM probe, respectively. The data analysis was carried out as in our previously reported work (*Kruger et al., 2019*; *Bell et al., 2020*; *Kruger et al., 2020*; *McGowan et al., 2020*).

## Pharmacological inhibition of Myosin in *Drosophila* follicles

Flies were fed wet yeast paste for 2–3 days and maintained at room temperature. Ovaries of wild-type (*yw*) or *fascin* mutant (*fascin^sn28/sn28*) flies were then dissected in S9 medium (*Prasad and Montell, 2007*), as described above. Ovarioles were teased apart and then were incubated at room temperature for 2 hr in either control media (S9 media + vehicle (DMSO)) or 200 µM of Y-27632. After 2 hr, ovaries were rinsed three times with S9 media and then fixed and stained following the pMRLC staining protocol described above.

## Western blot

Approximately 100 S9 follicles were dissected in room temperature Grace's insect media (Lonza, Walkersville, MD, USA or Thermo Fisher Scientific, Waltham, MA) and transferred to a 1.5 mL microcentrifuge tube containing 50 µL of Grace's media. Grace's media was removed and replaced with 50 µL 1 x PBS, 50 µL 2 X SDS Sample Buffer was added and the tissue lysed by grinding with a plastic pestle. Ten µL of sample were loaded per lane on either 8 % or 10 % SDS-PAGE gels. Western blots were performed using standard methods. The membranes were cut prior to incubation with primary antibodies to allow for two proteins to be simultaneously assessed. The following primary antibodies were used: rat α–Vasa Spradling, A.C.; obtained from the Developmental Studies Hybridoma Bank (DSHB), 1:100 and rabbit α-Zipper (Karess, R.; Institut Jacques Monod, Paris, France; *Wheatley et al., 1995*), 1:10,000. For the Vasa primary antibody incubations, the antibody was diluted in 5 % non-fat milk in 1 x Tris-buffered saline and 0.1 % Tween-20. For the Zipper primary antibody incubations, the antibody was diluted in 5 % Bovine Serum Albumin in 1 x Tris-buffered saline and 0.1 % Tween-20. The following secondary antibodies were used: Peroxidase-AffiniPure Goat Anti-Rat IgG (H + L), 1:5000 and Peroxidase-AffiniPure Goat Anti-Rabbit IgG (H + L), 1:10,000 (Jackson ImmunoResearch Laboratories, West Grove, PA, USA). Blots were developed with SuperSignal West Pico or Femto Chemiluminescent Substrate (Thermo Scientific, Waltham, MA, USA) and imaged using the Amersham Imager 600 (GE Healthcare Life Sciences, Chicago, IL). Bands were quantified using densitometry analysis in ImageJ (*Abramoff et al., 2004*). Zipper levels were assessed using four independent, biological samples per genotype, and statistical significance was determined using a two-sample t-test with unequal variance in Excel (Microsoft, Redmond, WA, USA).

## Acknowledgements

We thank the Westside Fly Group and Dunnwald lab for helpful discussions, and the Tootle lab for helpful discussions and careful review of the manuscript. Stocks obtained from the Bloomington *Drosophila* Stock Center (NIH P40OD018537) were used in this study. Information Technology Services – Research Services provided data storage support. This project is supported by National Institutes of Health R01GM116885. MCL was partially supported by the University of Iowa Summer Graduate Fellowship.

## Additional information

### Funding

| Funder | Grant reference number | Author |
| --- | --- | --- |
| National Institute of General Medical Sciences | R01GM116885 | Maureen C Lamb<br>Samuel Q Mellentine<br>Tina L Tootle |

The funders had no role in study design, data collection and interpretation, or the decision to submit the work for publication.

### Author contributions

Maureen C Lamb, Conceptualization, Formal analysis, Investigation, Methodology, Visualization, Writing - original draft; Chathuri P Kaluarachchi, Formal analysis, Investigation, Writing - review and editing; Thiranjeewa I Lansakara, Formal analysis, Investigation, Methodology; Samuel Q Mellentine, Formal analysis, Investigation; Yiling Lan, Formal analysis; Alexei V Tivanski, Supervision, Writing - review and editing; Tina L Tootle, Conceptualization, Funding acquisition, Supervision, Writing - review and editing

### Author ORCIDs

Maureen C Lamb http://orcid.org/0000-0002-4522-1910
Chathuri P Kaluarachchi http://orcid.org/0000-0003-2538-3952
Tina L Tootle http://orcid.org/0000-0002-1515-9538

### Decision letter and Author response

Decision letter https://doi.org/10.7554/eLife.69836.sa1
Author response https://doi.org/10.7554/eLife.69836.sa2

## Additional files

### Supplementary files

• Transparent reporting form

### Data availability

All data generated or analysed during this study are included in the manuscript as Figure-specific source data files.

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
