## [Decision Letter]

**Acceptance summary:**

Using the *Drosophila* border cell migration, a model collective cell migration, Lamb, Tootle and colleagues found that the actin bundling protein fascin has an antagonistic relationship non-muscle myosin II. By eliminating fascin expression in border cells, non-muscle myosin II activity increased. However, altering border cell contractility resulted in changes in the stiffness of the surrounding substrate demonstrating that migrating cells are not simply responding to the stiffness of their environment in vivo, but are actively modulating it. This manuscript will be of interest to the actomyosin and cell migration field, and more broadly to developmental and cell biologists.

**Decision letter after peer review:**

Thank you for submitting your article "Fascin limits Myosin activity within *Drosophila* border cells to control substrate stiffness and promote migration" for consideration by *eLife*. Your article has been reviewed by 3 peer reviewers, including Derek Applewhite as Reviewing Editor and Reviewer #1, and the evaluation has been overseen by Anna Akhmanova as the Senior Editor. The following individual involved in review of your submission has agreed to reveal their identity: Jordan Beach (Reviewer #3).

Essential revisions:

1) I would encourage the authors to limit assumptions of prior knowledge of fly development and border cell migration terminology. The authors make some efforts to guide the reader along at points (cartoon in figure 2A), but even these could use some additional information. I will note a few specific points, but I would recommend that for a journal with broad readership like this, a more guided tour of the research, especially the genetics, would be helpful:

– Figure 2A: border cells are not labeled. Follicle cells are not labeled. Why are follicle cells green and the data in Figure 2D also green? This is a little confusing. Related here: it took us a while to realize that "for increased clarity, graphs quantifying stiffness are represented in green" meant that throughout the paper, those graphs were in green. And this is true for other measurements as well. Perhaps the Y axis could also be colored to make this clear and "throughout the manuscript" could be added the first time this idea is conveyed? So "Stiffness, kPa" in green and "Nurse cell pMRLC" in blue? I'm not wed to the y-axis colorization, but some clarity on the coloring throughout the manuscript is advised because we both missed it the first couple times through.

– A cartoon guide to which cells are expected to lose fascin in germline, somatic, border cell knockdown (Figure 5) would be helpful. Perhaps a similar cartoon to figure 2A

2) The authors should more explicitly indicate how they measured p-MRLC levels in border cells versus nurse cells. How p-MRLC "puncta" are measured, and in particular what the authors mean by "length" of puncta, should be clarified. More notably, the p-MRLC staining looks quite different from the MRLC-GFP images shown.

3) The authors should clarify how many stage 9 follicles (egg chambers) they measured in each AFM experiment and for each genotype. In the Materials and methods, it says that 2-3 follicles were measured for each experiment. This seems like a low number, although it is a technically challenging method. A recent study from the Bilder lab (Chen et al., Nature Communications 2019) appeared to measure at least 8 follicles per genotype. This is particularly important, since the data points for the stiffness measurements are generally quite broad and overlap between controls and mutants, e.g., with ~5-15 kPa in control nurse cells and ~15-45 kPa in fascin null nurse cells (e.g., Figure 2D; but also Figure 5G).

4) Third, the non-autonomous control of nurse cell substrate stiffness, and levels of activated myosin in nurse cells, by loss of fascin in border cells (and by overexpression of activated Rho-kinase in border cells) is interesting and novel. The authors propose that the border cells regulate the stiffness of nurse cells to facilitate border cell migration. Further clarification of this phenotype would strengthen the manuscript. Specifically, do the authors find elevated p-MRLC in nurse cells that are in front of the border cells, or a more general elevation of p-MRLC levels (and presumably nurse cell stiffness)?

5) The authors use pharmacological inhibition of myosin and/or activation of myosin to rescue border cell migration (Figure 3 and Figure 3, figure supplement 1). The Y-27632 drug and MRLC-RNAi should be fine. However, *Drosophila* myosin has been reported to be insensitive to blebbistatin (Straight et al., Science 2003; Heissler et al., FASEB J. 2015). Therefore, caution should be taken in assessing the results with blebbistatin in *Drosophila*.

6) In Figure 3, the authors state that they were unable to knock down sqh by RNAi in border cells. Mishra et al. (Mol Biol Cell 2019) drove sqh RNAi with c306-GAL4 along with temperature-sensitive GAL80 to bypass lethality. This may be a way to decrease myosin levels just in fascin mutant border cells. Alternatively, the authors could overexpress constitutively activated Mbs (myosin phosphatase; Mbs N300), which should similarly reduce myosin activation (Lee and Treisman, Mol Biol Cell 2004).

7) With respect to the influence of the border cells on nurse cell stiffness – Do fascin mutant border cells mainly increase stiffness in front of the cluster, behind the cluster, or everywhere? While it would be difficult to measure stiffness using AFM in this case, the authors can examine p-MRLC. What happens when only some border cells are mutant for fascin (and/or overexpress activated Rok)? In this case, does it change which nurse cells have elevated p-MRLC?

8) p-MRLC immunostaining is used throughout and normalized to phalloidin staining or "background staining in the same follicle". We have a couple of concerns here. If you are down regulating a key actin bundling protein, should you be normalizing p-MRLC to actin? Couldn't F-actin be going down and p-MRLC stay the same, giving a relative increase in p-MRL:Actin ratio? Second, what is "background staining" in a follicle? Myosin is expressed in all of these cells and will be present in both the cortex and cytoplasm. Any "background" will be a combination of noise and real myosin signal. Where this background is taken is important. Traditionally, a phosphorylation event would be normalized to the same protein with that is being phosphorylated. Can the authors not normalize to MHC or MRLC? Similarly, are total levels of MHC/zipper and MRLC/sqh normal when manipulating fascin expression?

9) Can the authors further elaborate on how they think border cells influence nurse cell stiffness? Do the border cells "tug" on the nurse cells as they migrate, possibly through adhesion (and actomyosin) as border cells migrate upon the nurse cells? Possibly this may be clarified if the authors can analyze which nurse cells have elevated p-MRLC when border cells are mutant for fascin – in other words, is it only the nurse cell in front of the cluster with high p-MRLC, or is it all nurse cells?

10) Regarding novelty of cells controlling the stiffness of their substrate, I'm not convinced that this idea has not been entirely unexplored. The authors state 544- "Together our data uncover the transformative finding that collectively migrating cells modulate the stiffness of their substrate (Figure 7)". I think the novelty is more complex, considering the substrate here is another cell type. I'd note at least two recent papers that demonstrate similar ideas with cell:ECM interactions (below). I would encourage the authors to reserve some of their novelty claims for cell:cell migration, or clarify if we are misunderstanding their conclusions relative to the previous models. Also is there ECM in between these border and nurse cells?

– van Helvert and Friedl, 2016 (cited).

– Doyle, Yamada and colleagues, Dev Cell 2021 (not cited).

---

## [Author Response]

Essential revisions:1) I would encourage the authors to limit assumptions of prior knowledge of fly development and border cell migration terminology. The authors make some efforts to guide the reader along at points (cartoon in figure 2A), but even these could use some additional information. I will note a few specific points, but I would recommend that for a journal with broad readership like this, a more guided tour of the research, especially the genetics, would be helpful:– Figure 2A: border cells are not labeled. Follicle cells are not labeled. Why are follicle cells green and the data in Figure 2D also green? This is a little confusing. Related here: it took us a while to realize that "for increased clarity, graphs quantifying stiffness are represented in green" meant that throughout the paper, those graphs were in green. And this is true for other measurements as well. Perhaps the Y axis could also be colored to make this clear and "throughout the manuscript" could be added the first time this idea is conveyed? So "Stiffness, kPa" in green and "Nurse cell pMRLC" in blue? I'm not wed to the y-axis colorization, but some clarity on the coloring throughout the manuscript is advised because we both missed it the first couple times through.– A cartoon guide to which cells are expected to lose fascin in germline, somatic, border cell knockdown (Figure 5) would be helpful. Perhaps a similar cartoon to figure 2A

We added a new Figure 1 to provide more details and background on *Drosophila* oogenesis and border cell migration. Further, the colors of the y-axes labels now match the data points on the graphs, and where appropriate, reflect the colors in the follicle diagrams. Additionally, text has been added to the Results to clarify the relationship between the colors and types of quantifications. We have also added a follicle schematic to Figure 6 (previously Figure 5) to clarify where RNAi knockdown of Fascin is occurring.

2) The authors should more explicitly indicate how they measured p-MRLC levels in border cells versus nurse cells. How p-MRLC "puncta" are measured, and in particular what the authors mean by "length" of puncta, should be clarified. More notably, the p-MRLC staining looks quite different from the MRLC-GFP images shown.

A supplemental figure (Figure 2 – supplemental figure 1) has been added to describe in detail how the p-MRLC measurements were performed in the border cells and nurse cells and to provide examples. Additional control data and experiments are also provided in this supplemental figure. Brief descriptions have also been added to the Results section to describe each of the quantifications, including those of the puncta. In the Discussion, we address the differences in the pMRLC staining and the MRLC-GFP imaging.

3) The authors should clarify how many stage 9 follicles (egg chambers) they measured in each AFM experiment and for each genotype. In the Materials and methods, it says that 2-3 follicles were measured for each experiment. This seems like a low number, although it is a technically challenging method. A recent study from the Bilder lab (Chen et al., Nature Communications 2019) appeared to measure at least 8 follicles per genotype. This is particularly important, since the data points for the stiffness measurements are generally quite broad and overlap between controls and mutants, e.g., with ~5-15 kPa in control nurse cells and ~15-45 kPa in fascin null nurse cells (e.g., Figure 2D; but also Figure 5G).

We apologize for not being clear in the methods and thank the reviewers for pointing this out. We originally stated that 2-3 follicles were measured for each experiment but failed to indicate the experiments were performed 3 times. We have corrected this in the Materials and methods, and now state that 2-3 follicles were measured *per experiment* for a total of 3 experiments (6-9 follicles total per genotype). Additionally, the n values for each genotype have been added to the graphs.

4) Third, the non-autonomous control of nurse cell substrate stiffness, and levels of activated myosin in nurse cells, by loss of fascin in border cells (and by overexpression of activated Rho-kinase in border cells) is interesting and novel. The authors propose that the border cells regulate the stiffness of nurse cells to facilitate border cell migration. Further clarification of this phenotype would strengthen the manuscript. Specifically, do the authors find elevated p-MRLC in nurse cells that are in front of the border cells, or a more general elevation of p-MRLC levels (and presumably nurse cell stiffness)?

We have revisited our data and find that we can only make a qualitative statement about the spatial distribution of activated Myosin on the nurse cell membranes surrounding the border cell cluster. We find that activated Myosin (pMRLC) is observed in the nurse cells in all directions from the border cell cluster. Further, we observe more of a general elevation of pMRLC levels rather than a change in the spatial distribution in the *fascin*-null follicles. We have added this qualitative information to the Results and the Discussion. We also note in the Discussion that further characterization of this is warranted with more precise tools.

5) The authors use pharmacological inhibition of myosin and/or activation of myosin to rescue border cell migration (Figure 3 and Figure 3, figure supplement 1). The Y-27632 drug and MRLC-RNAi should be fine. However, *Drosophila* myosin has been reported to be insensitive to blebbistatin (Straight et al., Science 2003; Heissler et al., FASEB J. 2015). Therefore, caution should be taken in assessing the results with blebbistatin in Drosophila.

We thank the reviewers for pointing this out and have removed the blebbistatin data from our results.

6) In Figure 3, the authors state that they were unable to knock down sqh by RNAi in border cells. Mishra et al. (Mol Biol Cell 2019) drove sqh RNAi with c306-GAL4 along with temperature-sensitive GAL80 to bypass lethality. This may be a way to decrease myosin levels just in fascin mutant border cells. Alternatively, the authors could overexpress constitutively activated Mbs (myosin phosphatase; Mbs N300), which should similarly reduce myosin activation (Lee and Treisman, Mol Biol Cell 2004).

We have attempted to make a fly stock with *fascin* mutant, c306-GAL4, and temperature sensitive GAL80, but have been unsuccessful. The crosses necessary to generate this stock are complex since the *fascin* gene and c306-GAL4 driver are both on the X chromosome (requiring a recombinant chromosome that likes to separate) and the *fascin*-null females are sterile. Thus far we have been unable to generate these flies.

As an alternative we have attempted to knockdown Rok by RNAi in the border cells (c306-GAL4) of *fascin* mutants. We were able to obtain adult flies of *c306-GAL4, fascin^sn28^/fascin^sn28^;;UAS Rok RNAi*. However, moving these flies to 29ºC to drive knockdown resulted in most of the flies dying. Our limited analysis of the few remaining flies reveals that even in the control – *c306-GAL4; UAS Rok RNAi* – Myosin activity levels are not significantly reduced, suggesting that Rok is not sufficiently knocked down.

We have also been attempting to generate the *fascin^sn28^/FM7;; UAS Mbs-N300* stock to attempt this experiment another way. To generate this stock, we first had to make double balanced stocks on each chromosome. This line is just now being established, and therefore, we have not yet been able to attempt this experiment. We did, however, generate *c306 GAL4; UAS Mbs-N300* adult females. Initial examination of pMRLC levels reveals that expression of Mbs-N300 is not sufficient to significantly reduce pMRLC levels compared to controls; this finding suggests the proposed experiment in the *fascin* mutant background is unlikely to yield interpretable results.

7) With respect to the influence of the border cells on nurse cell stiffness – Do fascin mutant border cells mainly increase stiffness in front of the cluster, behind the cluster, or everywhere? While it would be difficult to measure stiffness using AFM in this case, the authors can examine p-MRLC. What happens when only some border cells are mutant for fascin (and/or overexpress activated Rok)? In this case, does it change which nurse cells have elevated p-MRLC?

These are very interesting questions. In relation to the spatial regulation of stiffening, the pMRLC data suggests that it extends from the border cells in all directions. How that is occurring temporally remains to be determined. Such information requires the development of tools that differentially label Myosin activity on the border cells from that of the nurse cells, and rapid, non-photobleaching imaging. We are currently attempting to build such tools.

In relation to what happens if some cells are mutant for *fascin*, that is an interesting question. We speculate that our current tools do not provide the resolving power to address this, as we expect that altering Myosin activity in one cell of the cluster, will push/pull on the other cells, driving rapid changes in Myosin activity in the wild-type cells. This would be particularly concerning for the Rok-CAT expression. Further, we are concerned about whether the experimental approach would work well enough to generate interpretable data. Specifically, while the experiment could be done using RNAi knockdown in a clonal manner, we are concerned whether the data will be interpretable as Fascin knockdown can be variable due to the high level of expression in the border cells. Thus, we think it would be better to look at mutant clones, however, no allele of *fascin* is currently available on an FRT chromosome. In summary, these are interesting questions that we hope to address in the future when we have developed the tools and imaging methods to resolve Myosin activity temporally and spatially.

8) p-MRLC immunostaining is used throughout and normalized to phalloidin staining or "background staining in the same follicle". We have a couple of concerns here. If you are down regulating a key actin bundling protein, should you be normalizing p-MRLC to actin? Couldn't F-actin be going down and p-MRLC stay the same, giving a relative increase in p-MRL:Actin ratio? Second, what is "background staining" in a follicle? Myosin is expressed in all of these cells and will be present in both the cortex and cytoplasm. Any "background" will be a combination of noise and real myosin signal. Where this background is taken is important. Traditionally, a phosphorylation event would be normalized to the same protein with that is being phosphorylated. Can the authors not normalize to MHC or MRLC? Similarly, are total levels of MHC/zipper and MRLC/sqh normal when manipulating fascin expression?

We have added a supplemental figure (Figure 2 – supplemental figure 1) to address these concerns. First, we depict how the measurements and calculations were made. We also show that Actin levels on the nurse cell membranes in wild-type and *fascin* mutant follicles are not significantly different from each other. Further, we clarified in the Results and Methods that “background staining” refers to the cytoplasm of the nurse cell. We also show that the levels of background pMRLC staining in wild-type and *fascin* mutant follicles are not significantly different. We were unable to use unphosphorylated Myosin as a control because of the lack of antibodies to *Drosophila* Myosin components, and the available antibodies for Myosin heavy chain (*Drosophila* Zipper) or MRLC (mammalian MRLCs) failed to work in our hands for immunofluorescence. We also now show that the protein level of Myosin heavy chain (Zipper) does not change between wild-type and *fascin*-null Stage 9 follicles by western blot (Figure 2 – supplemental figure 1).

9) Can the authors further elaborate on how they think border cells influence nurse cell stiffness? Do the border cells "tug" on the nurse cells as they migrate, possibly through adhesion (and actomyosin) as border cells migrate upon the nurse cells? Possibly this may be clarified if the authors can analyze which nurse cells have elevated p-MRLC when border cells are mutant for fascin – in other words, is it only the nurse cell in front of the cluster with high p-MRLC, or is it all nurse cells?

As mentioned above in Comment 4, we can only make a qualitative statement about the spatial distribution of activated Myosin on the nurse cell membranes surrounding the border cell cluster. Specifically, we find that activated Myosin is observed in the nurse cells in all directions from the border cell cluster in both wild-type and *fascin* mutant follicles. In the *fascin* mutants there is a general elevation of pMRLC levels across the nurse cells. We have added this qualitative information to the Results and the Discussion. In the Discussion, we point out that further characterization of the temporal and spatial regulation of Myosin activation is warranted and speculate on how the border cells influence nurse cell stiffness.

10) Regarding novelty of cells controlling the stiffness of their substrate, I'm not convinced that this idea has not been entirely unexplored. The authors state 544- "Together our data uncover the transformative finding that collectively migrating cells modulate the stiffness of their substrate (Figure 7)". I think the novelty is more complex, considering the substrate here is another cell type. I'd note at least two recent papers that demonstrate similar ideas with cell:ECM interactions (below). I would encourage the authors to reserve some of their novelty claims for cell:cell migration, or clarify if we are misunderstanding their conclusions relative to the previous models. Also is there ECM in between these border and nurse cells?– van Helvert and Friedl, 2016 (cited).– Doyle, Yamada and colleagues, Dev Cell 2021 (not cited).

We thank the reviewers for pointing this out. We have tempered our language about the novelty of our findings throughout the manuscript. In the discussion, we have added a paragraph discussing these and other references that support migrating cells influence their environment and explain how our findings extend that knowledge. In particular, we are studying a cell-on-cell migration in a native context.

We also have added text in both the introduction and the discussion clarifying that there is only a small punctum of ECM at one spot between the border cells and the nurse cells, and this ECM is not thought to serve as the substrate for migration.